# Metabolic dysregulation impairs lymphocyte function during severe SARS-CoV-2 infection

Sanjeev Gurshaney[1], Anamaria Morales-Alvarez [1], Kevin Ezhakunnel [1], Andrew Manalo[1], Thien-Huong Huynh[1], Jun-Ichi Abe [2], Nhat-Tu Le[3], Daniela Weiskopf[4], Alessandro Sette [4,5], Daniel S. Lupu[6], Stephen J. Gardell[7] & Hung Nguyen [1✉]

Cellular metabolic dysregulation is a consequence of SARS-CoV-2 infection that is a key determinant of disease severity. However, how metabolic perturbations influence immunological function during COVID-19 remains unclear. Here, using a combination of high-dimensional flow cytometry, cutting-edge single-cell metabolomics, and re-analysis of single-cell transcriptomic data, we demonstrate a global hypoxia-linked metabolic switch from fatty acid oxidation and mitochondrial respiration towards anaerobic, glucose-dependent metabolism in CD8+Tc, NKT, and epithelial cells. Consequently, we found that a strong dysregulation in immunometabolism was tied to increased cellular exhaustion, attenuated effector function, and impaired memory differentiation. Pharmacological inhibition of mitophagy with mdivi-1 reduced excess glucose metabolism, resulting in enhanced generation of SARS-CoV-2- specific CD8+Tc, increased cytokine secretion, and augmented memory cell proliferation. Taken together, our study provides critical insight regarding the cellular mechanisms underlying the effect of SARS-CoV-2 infection on host immune cell metabolism, and highlights immunometabolism as a promising therapeutic target for COVID-19 treatment.

[1] Cancer Division, Burnett School of Biomedical Science, College of Medicine, University of Central Florida, Orlando, FL 32827, USA. [2] Department of Cardiology, The University of Texas MD Anderson Cancer Center, 1400 Pressler Street, Unit 1451, Houston, TX 77030, USA. [3] Center for Cardiovascular Regeneration, Department of Cardiovascular Sciences, Houston Methodist Research Institute, Houston, TX 77030, USA. [4] Center for Infectious Disease and Vaccine Research, La Jolla Institute for Immunology (LJI), La Jolla, CA 92037, USA. [5] Department of Medicine, Division of Infectious Diseases and Global Public Health, University of California, San Diego (UCSD), La Jolla, CA 92037, USA. [6] AdventHealth Cancer Institute, AdventHealth Research Institute, Orlando, FL 32804, USA. [7] Translational Research Institute, AdventHealth Research Institute, Orlando, FL 32804, USA. ✉email: hung.nguyen@ucf.edu

Coronavirus disease 2019 (COVID-19) is one of the most severe health crises in history[1]. In the vast majority of infected individuals, the host immune response is sufficient to clear the infection. However, as a result of the insufficient and dysfunctional immune response towards SARS-CoV-2 infection, some individuals acquire severe disease marked by tremendous lung damage[2]. The growing body of evidence is suggestive of a link between host immune cells and patient metabolism during severe COVID-19[3]. Prior metabolic comorbidities and their associated cluster of conditions pose potent risk factors for disease severity[4–7]. Patients with type 2 diabetes, obesity, hyperglycemia, dyslipidemia, and older age have higher rates of severe complications and mortality[8]. Interestingly, host metabolic rebalance using cholesterol-lowering-(statins), glucose metabolism-reducing (2-deoxyglucose), and antioxidant (melatonin) drugs has been shown to benefit COVID-19 treatment[9].

T cells (Tc) including CD4 and CD8 are major cell subsets providing protective immunity against SARS-CoV-2 infection[10]. Indeed, SARS-CoV-2- induced Tc lymphopenia is evident in 83.2% of COVID-19 patients with acute respiratory distress syndrome (ARDS)[11]. The processes of Tc activation, differentiation, and maturation into memory and effector subsets are tightly regulated by metabolic reprogramming[12]. Upon Tc receptor (TCR) activation, mTOR signaling initiates a glucose transporter 1 (glut-1)- mediated increase in aerobic glycolytic flux that is required for activation[13]. Whereas the downstream differentiation of effector CD8+Tc is heavily dependent on the activity of prolonged aerobic glycolysis; increased lipid uptake and mitochondrial fatty acid oxidation (FAO) are required for memory cell differentiation[14]. It has been shown that under hypoxic conditions, hypoxia-inducible factor a (HIF-1a) redirects pyruvate from mitochondrial shuttling towards lactate conversion while increasing Tc intrinsic reactive oxidative species (ROS), resulting in impaired mitochondrial function and hampered memory cell differentiation[5]. While the accumulation of ROS was found in Tc from COVID-19 patients with ADRS[9], it is completely unknown how the condition of oxygen deprivation would affect the balance of effector and memory Tc phenotypes as well as the abundance of viral antigen-specific Tc during COVID-19. Moreover, it is also unclear whether Tc fate and function would be different in severe COVID-19 compared to other lung pathologies.

Mitophagy is a cellular process involved in the selective degradation of damaged mitochondria[15]. During conditions of hypoxia, cells upregulate mitophagy to direct their metabolism towards aerobic glycolysis as a mechanism to maintain their survival[16]. While enhancing mitochondrial damage has been identified in Tc from COVID-19 patients with ADRS[5], the role of mitophagy in regulating mitochondrial functions during SARS-CoV-2 infection remains unexplored. Importantly, inhibition of mitophagy has been shown to inhibit viral proliferation in virally infected cells; thus, suggesting that mitophagy-targeting strategies may have a combinatorial effect of altering host immunometabolism as well as directly decreasing viral load[17].

The abundance of CD56+CD8+ cells (NKT) is a strong predictive biomarker for COVID-19 outcome[18]. It has been apparent that NKT plays an important role in the prevention of COVID-19-induced pneumonia[19]. As a key bridge between innate and adaptive immunity[20], little is known about how NKT metabolism during SARS-CoV-2 infection. Interestingly, in contrast to Tc, NKTs are considerably more dependent on mitochondrial metabolism after activation[21,22]. Because mitochondrial dysfunction is identified in the majority of immune cells during SARS-CoV-2 infection[9], understanding how ADRS-induced hypoxia affects NKT function in COVID-19 is critically important.

The lung is the primary target organ of SARS-CoV-2, as the spike protein directly binds to ACE2 receptors expressed on the surface of lung epithelial cells (ECs)[23]. As a result, severe COVID-19 is characterized by profound lung damage, resulting in decreased blood oxygen saturation (hypoxia), as well as increased serum lactate dehydrogenase (LDHA) level[2,24,25]. Both downstream hypoxia signaling and hyperlactatemia have been associated with pro-inflammatory cytokine syndrome and lymphocyte dysfunction[26,27]. However, it is not completely understood how hypoxia in COVID-19 patients affects the metabolic phenotype of Tc through attenuating EC function in the lung of COVID-19 patients.

In the current study, using high-dimensional flow cytometry and cutting-edge single-cell metabolomics of PBMCs from hospitalized COVID-19 patients, we demonstrated that metabolic disorders by hypoxia and anaerobic glycolysis induced dysfunctional CD8+Tc and NKTs during SARS-CoV-2 infection. We show an impaired, hypoxia triggered memory cell differentiation in CD8+Tc of COVID-19 patients. Finally, mitophagy was found to be an important regulator of immunometabolic function in CD8+Tc and ECs. Intriguingly, pharmacological inhibition of mitophagy via mdivi-1 enhanced effector function as well as rescued memory differentiation function amongst CD8+Tc. Publicly available single-cell sequencing datasets on the bronchoalveolar lavage fluid (BALF) and PBMCs from COVID-19 patients were reanalyzed to validate the metabolic reprogramming in CD8+Tc, NKTs, and ECs at transcriptomic level. Altogether, the current study provides key, insightful cellular and molecular mechanisms underlying a critical link between lung dysfunction, metabolic dysregulation, and impaired lymphocyte function during SARS-CoV-2 infection.

## Results

**High-dimensional immunophenotyping reveals a distinct COVID-19 immunophenotype in both circulation and the BALF.** COVID-19 patients with severe disease have been found to suffer from substantial immune dysregulation[28,29]. We first performed high-dimensional immunophenotyping of peripheral blood mononuclear cells (PBMCs) from non-infected controls (healthy), hospitalized COVID-19 patients (COVID(+)), as well as from non-COVID (by PCR test) patients with COVID-like upper respiratory symptoms requiring intensive care, abbreviated as COVID(−). (Fig. 1a, Supplementary Fig. 1, Supplementary Table 1,5). Analysis was performed on freshly isolated cells without cryogenic preservation to best reflect the metabolic/functional state of cells in the body. Comparative evaluation of patient samples by principal component analysis (PCA) revealed distinct clustering of healthy, COVID(−), and COVID(+) patients, suggesting an abnormal immunophenotype of PBMCs during SARS-COV-2 infection (Fig. 1b). CD8+Tc are the main cellular immune population that governs viral clearance[30,31]. Corroborating prior reports, high-dimensional flow cytometry (Supplementary Fig. 2A–C) revealed significant lymphopenia of multiple CD8- derived subsets in PBMCs from hospitalized COVID(+) patients (Fig. 1c–e). Bulk CD8+Tc were dramatically decreased in COVID(+) patients compared to both COVID(−) and healthy patients (Fig. 1e). The percentage of circulating effector CD8+Tc was increased in COVID(+) patients compared to healthy donors; however, significantly decreased compared to that of COVID(−) patients (Fig. 1e). Additionally, the percentage of circulating CD8+T_M was heavily reduced in severe COVID(+) patients as compared to healthy controls (Fig. 1d, g). However, this consistent decrease was not noted amongst COVID(−) patients, indicating impaired memory differentiation specifically occurs in SARS-CoV-2 infection (Fig. 1d, g). Given that the lung

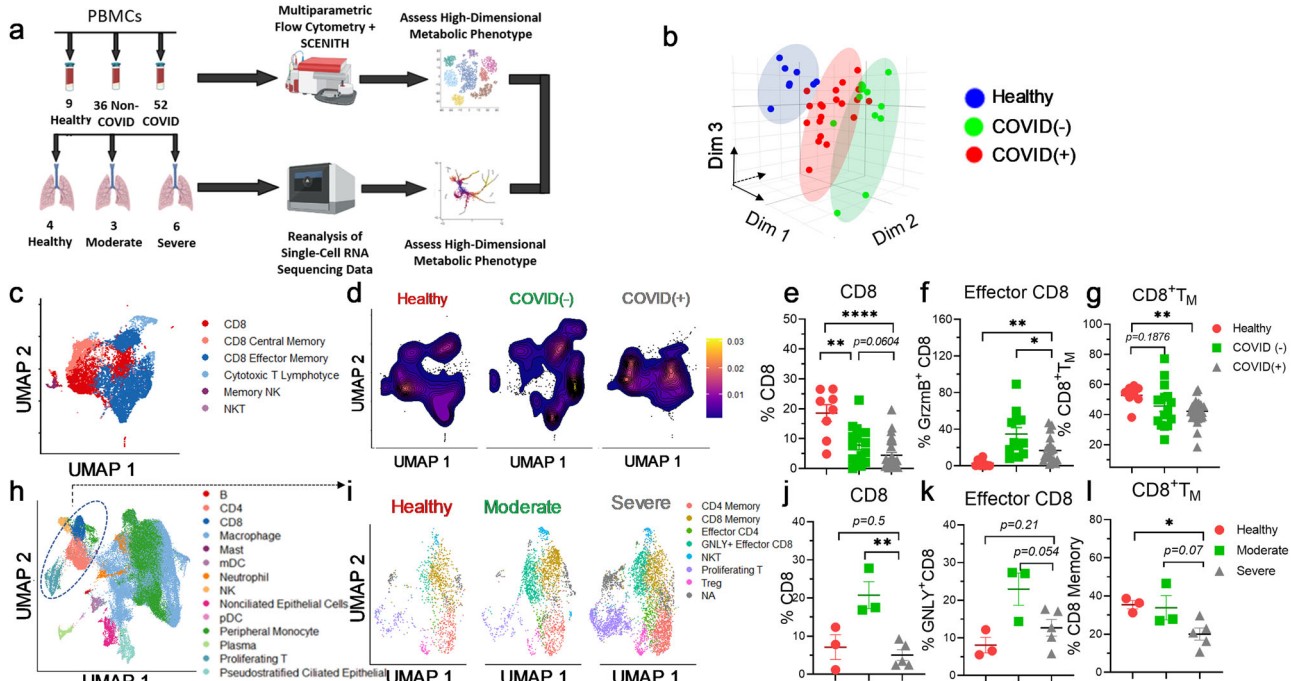

**Fig. 1 Distinct immunophenotype of BALFs and PBPMCs from COVID-19 patients. a** Schematic illustrating experimental design for multiparametric flow cytometry and single-cell RNA-sequencing re-analysis. **b–g** The total PBMC patient cohort includes 9 healthy, 36 COVID(–), and 52 COVID-19+ patients; given the high degree of lymphopenia, various subsets of the total cohort were used for different experiments. **h–l** The total BALF patient cohort includes 4 healthy, 3 moderate COVID-19, and 6 severe COVID-19 patients. One healthy and one severe patient was excluded from downstream analysis due to low T-cell count. **b** 3D PCA analysis conducted using bulk expression of each marker per sample as input; circles were manually drawn around the PCA plot to highlight distinct clustering. **c** UMAP projection of labeled PBMC populations from 8 Healthy (HD), 12 COVID (−) (C−), and 17 COVID (+) (C+) patients. **d** Representative contoured kernel density for UMAP projection of PBMCs, a representative sample from each group was displayed. **e–g** Summary graphs demonstrating frequency of CD8+Tc amongst all live cells in patient PBMCs ($n = 8$ HD,15 COVID (−), 27 C+). **e** frequency of effector CD8Tc (grzmB+ CD8Tc) amongst all CD8+Tc ($n = 8$ HD, 12 C−, 20 C+) (**f**), and frequency of CD8+Tc memory (CD8+T_M) amongst all CD8+Tc ($n = 8$ HD, 16 C−, 26 C+) (**g**). **h** UMAP projection displaying population labeling of 66,452 cells from healthy (HD), moderate (M), and severe (S) COVID-19+ patients, Tc populations were circled manually. **i** UMAP projections displaying labeled unsupervised clustering analysis of 7601 reintegrated cells split between healthy, moderate, and severe COVID-19+ patients. **j–l** Summary graphs demonstrating frequency of bulk CD8+Tc amongst total BALF cells ($n = 3$ HD, 3 M, 5 S) (**j**), effector CD8+Tc among all Tc ($n = 3$ HD, 3 M, 5 S) (**k**), and CD8+Tc memory (CD8+T_M) ($n = 3$ HD, 3 Moderate (M), 5 Severe (S), (**l**) amongst all Tc. Two-tailed student's t test was used. *$p < 0.05$, **$p < 0.01$, and ***$p < 0.001$. Biorender was used in part to create the schematic in **a**; appropriate license and permission to use figure in publication were obtained.

is the primary target for COVID-19 attack[1], we examined whether CD8+Tc lymphopenia could also be detected in bronchoalveolar lavage fluid (BALF) cells. We reanalyzed a publicly available single-cell RNA-sequencing (scRNA-seq) dataset of BALF samples from COVID(+) patients[32]. Tc were identified (Supplementary Fig. 3a, b), subsetted, and a second round of unsupervised clustering and UMAP dimensionality reduction was performed to identify and delineate distinct, clear Tc subsets (Fig. 1h, l, Supplementary Fig. 4, and Supplementary Table 2). Similarly, effector and bulk CD8+Tc were significantly increased in the BALF of moderate COVID(+) patients compared to healthy controls (Fig. 1i–k). Noticeably, patients with moderate as compared to severe symptoms had dramatically higher abundance of effector CD8+Tc in the BALF (Fig. 1i, k). Finally, we observed an attenuation in the percentage of CD8+T_M in COVID-19(+) BALFs, which was most apparent in severe patients (Fig. 1i, l), corroborating the impaired memory cell abundance found in PBMCs (Fig. 1d, g).

**CD8+Tc metabolically reprogram towards glycolytic dependence and exhibit impaired mitochondrial function during SARS-CoV-2 infection.** Metabolic dysregulation is well-recognized in COVID-19 pathogenicity and has been linked to a dysfunctional immune response[33]. The function and

immunological fate of CD8+Tc specifically has been shown to be heavily dependent on metabolism during viral infection[8]. We assessed the immunometabolic profiles of CD8+Tcs on freshly isolated patient-derived PBMCs. Because hypoxia and excessive glycolysis are evident in COVID(+) patients with severe disease[25,34,35], we first examined the degree of glucose uptake in CD8+Tc during SARS-COV-2 infection. Glucose transporter 1 (glut-1) is a receptor that facilitates glucose uptake by Tc during viral infection[36]. Indeed, expression of glut-1 was augmented in CD8+Tc of COVID(+) patients (Fig. 2a). To assess the cellular dependence of COVID-19(+) CD8+Tc on glycolysis, we leveraged SCENITH single-cell metabolomics assay. COVID(+) CD8+Tc had increased glycolytic flux compared to healthy and COVID(−) patients (Fig. 2b). Accelerated glut1 expression and glycolytic flux in CD8+Tc from COVID-19(+) compared to those from COVID(−) patients (Fig. 2a, b) suggested dysfunctional CD8+T_C glucose metabolism as a hallmark of SARS-CoV-2 infection. Under hypoxic condition, excessive dependence on glycolytic flux has been linked to mitochondrial dysfunction[37]. As expected, decreased mitochondrial membrane potential, indicative of impaired mitochondrial function was selectively noted in COVID(+) CD8+Tc (Fig. 2c). Consistently, SCENITH metabolomics analysis revealed a strong reduction in the mitochondrial flux of COVID(+) CD8+Tc compared to those from both healthy

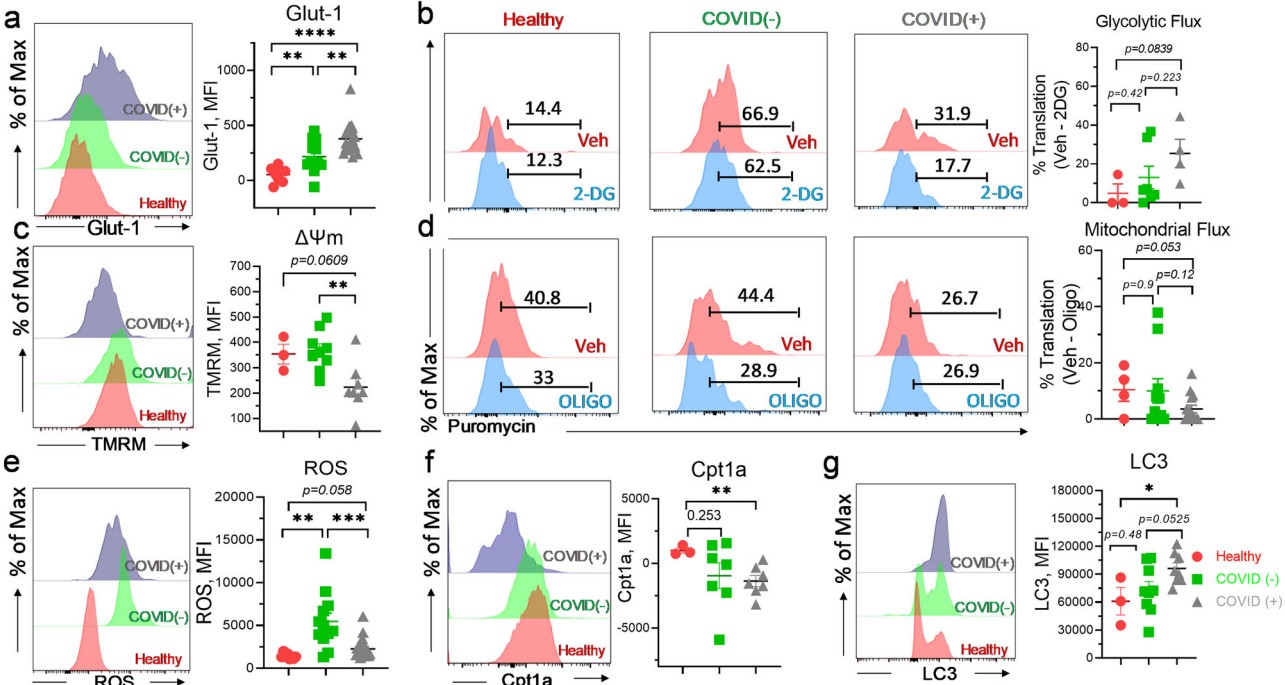

**Fig. 2 Metabolic dysfunction is evident in CD8+Tc from severe COVID-19 patients.** Freshly isolated PBMCs were evaluated by flow cytometry. **a** Representative histogram and summary graphs demonstrating glut-1 expression in CD8+Tc from healthy, COVID(−), and COVID(+) patients ($n = 8$ HD, 12 C−, 20 C+). **b** Representative histograms and summary graph demonstrating glycolytic flux of CD8+Tc from healthy, COVID(−), and COVID(+) patients using SCENITH ($n = 3$ HD, 7C−, 4 C+). **c** Representative histogram and summary graph demonstrating ΔΨm, mitochondrial membrane potential (TMRM), in CD8+Tc from healthy, COVID(−), and COVID(+) patients ($n = 3$ HD, 9 C−, 8 C+). **d** Representative histograms and summary graph demonstrating mitochondrial flux of CD8+Tc from healthy, COVID(−), and COVID(+) patients by SCENITH assay ($n = 4$ HD, 10 C−, 14 C+). **e–g** Representative histogram and summary graphs demonstrating the levels of ROS ($n = 8$ HD, 12 C−, 20 C+) (**e**), cpt1a ($n = 3$ HD, 7 C−, 7 C+) (**f**), and lc3 ($n = 3$ HD, 9 C−, 8 C+) **g** in CD8+Tc from healthy, COVID(−), and COVID(+) patients. **c, d, g** Three samples were excluded from the analysis due to critically low CD8+Tc count; in **b**, one sample was excluded from analysis due to critically low CD8+Tc count. Two-tailed student's t test was used. *$p < 0.05$, **$p < 0.01$, and ***$p < 0.001$.

and COVID(−) patients (Fig. 2d). Together, these results clearly indicated a functional metabolic switch from mitochondrial respiration to glucose-dependent metabolism in CD8+Tc during SARS-CoV-2 infection. Prolonged anaerobic and mitochondria-independent glycolysis was reported to impair reductive NADPH activity, resulting in the propagation of oxidative stress[38,39]. As predicted, an accumulation of ROS was detected in CD8+Tc from COVID-19 (+) patients (Fig. 2e). Mitochondrial FAO is an important mechanism for cells to prevent excessive oxidative stress production[40] and to generate acetyl-CoA (Ac-CoA) required maintaining stemness[41]. Indeed, decreased expression of carnitine palmitoyltransferase 1 A (cpt1a), an enzyme catalyzed the transfer of long-chain FA through mitochondrial membrane for subsequent oxidation[42], was also found indicating attenuated FAO in CD8+Tc from COVID-19 patients (Fig. 2f). Additionally, enhanced expression of LC3 (Fig. 2g) indicated prevalence of CD8+Tc autophagy in COVID-19. Along with mitochondrial impairment, these results demonstrated upregulation of mitophagy in CD8+Tc during SARS-CoV-2 infection.

**Metabolically linked exhaustion in CD8+Tc during severe SARS-CoV-2 infection.** Given that metabolism is a critical regulator of immune cell function[43], we next examined the functional characteristics of CD8+Tc during COVID-19. Augmented expression of surface glycoprotein lymphocyte activation gene-3 (lag-3) is indicative of increased cellular exhaustion of CD8+Tc in COVID(+) patients (Fig. 3a). We further found that cellular protein synthesis, which is required for production of effector molecules and cytokines in activated Tc[44], was reduced in

COVID(+) as compared to COVID(−) CD8+Tc (Fig. 3b). In support of this observation, remarkable decrease in ki67 expression was detected in CD8+Tc from COVID(+) patients (Fig. 3c) validating an impairment in CD8+Tc proliferation during SARS-CoV-2 infection. Increased hif-1α was detected during severe SARS-CoV-2 infection as a result of reduced oxygen saturation and hypoxia[45,46]. Notably, there was a significantly increased expression of hif-1α in lag-3^high CD8+Tc from COVID(+) patients (Fig. 3d) suggesting that a hypoxia-mediated metabolic switch may implicate in CD8+Tc dysfunction. Along this line, COVID-19(+) CD8+Tc with lower mitochondrial mass exhibited impaired IFNγ secretion capacity (Fig. 3e).

To investigate whether immunometabolic dysregulation in CD8+Tc can be a potential mechanism underlining increased COVID-19 severity within patients with metabolic disorders, we evaluated whether COVID(+) patients in our cohort who had metabolic syndrome (see Methods for detailed classification) possessed differential immunometabolic profile. We observed a significant increase in the expression of hif-1α on CD8+Tc amongst COVID(+) patients with metabolic syndrome, accompanied by a mild increase in glut-1 expression, suggesting that patients with prior metabolic comorbidities may have an increased hypoxia-driven anaerobic glycolysis (Supplementary Fig. 5a). In patients who had elevated level of serum lactate dehydrogenase (LDH) (key glycolysis rate-limiting enzyme), we found a moderate increase in the expression of hif-1α and lag-3 (Supplementary Fig. 5b), further highlighting the potential connection between altered patient metabolism and dysregulated immune function during SARS-CoV-2 infection. Accordingly,

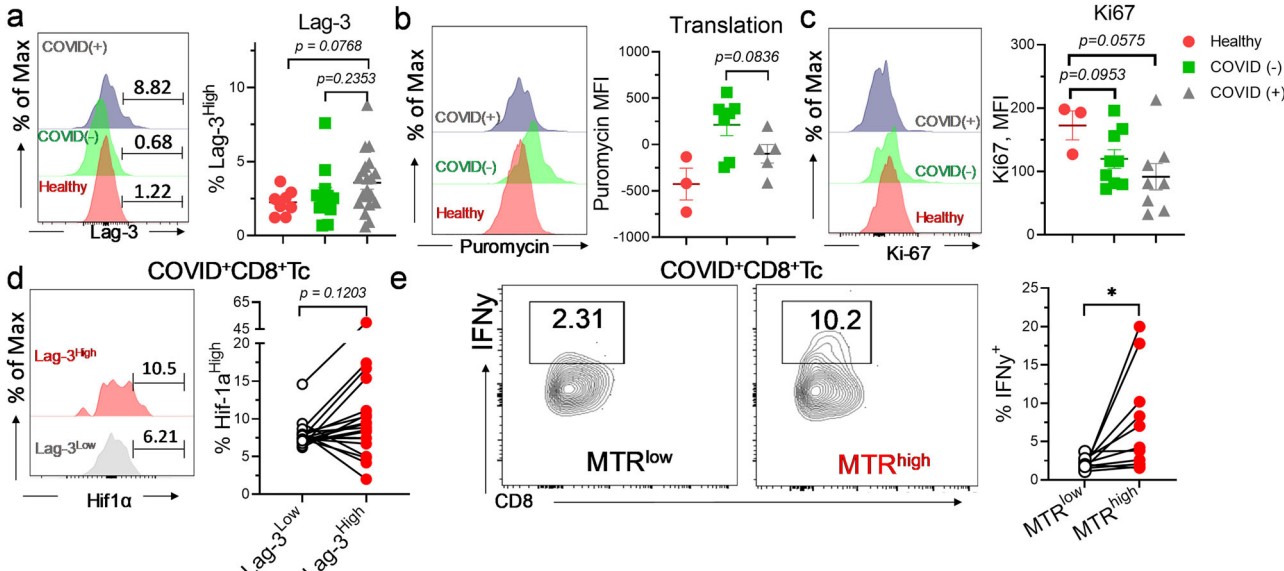

**Fig. 3 CD8+Tc are functionally exhausted in severe COVID-19 patients.** Freshly isolated PBMCs were evaluated for metabolic properties by flow cytometry. **a–c** Representative histogram and summary graph demonstrating the abundance of lag-3+ CD8+Tc (n = 8 HD, 12 C−, 20 C+) (**a**), cellular translational level via Puromycin (n = 3 HD, 7 C−, 5 C+) (**b**), and Ki-67 expression (n = 3 HD, 9 C−, 8 C+) (**c**) in CD8+Tc from healthy, COVID(−), and COVID(+) patients. **d** Representative histogram and paired graph abundance of HIF-1α^High cells in lag-3^low and lag-3^highCD8+Tc from COVID(+) patients (n = 18). **e** Representative contour plot and summary paired graph demonstrating IFNy+ cells in MTR^low and MTR^high CD8+Tc from COVID(+) patients (n = 11). **b**, **d** One sample was excluded due to critically low CD8+Tc count; in **c**, three samples were excluded due to critically low CD8+Tc count. Two-tailed student's t test was used. *p < 0.05, **p < 0.01, and ***p < 0.001.

positive correlations between serum glucose level and lag-3 or glut1 expression in CD8+Tc and between LDH and CD8+Tc expression of lag-3 and VDAC-1 were identified in COVID-19 patients (Supplementary Fig. 5c).

**Hypoxia-mediated metabolic reprogramming in COVID-19(+) CD8+Tc.** To better understand the transcriptional changes underlying the relationship between metabolic dysfunction and impaired immune cell function in the primary site of viral attack, we profiled the metabolic landscape of BALF effector CD8+Tc from COVID(+) patients. Differential expression analysis revealed increased expression of genes encoding anaerobic glycolysis (*GAPDH, GALM,* and *ALDOA*) in effector CD8+Tc from moderate and severe COVID(+) patients (Supplementary Fig. 6a, b). Key metabolic pathways including hypoxia, anaerobic glycolysis, mitophagy, autophagy, cell exhaustion, and senescence were upregulated, while pathways relying on mitochondrial metabolism including FAO, cholesterol metabolism, and oxidative phosphorylation (OXPHOS) were attenuated in effector CD8+Tc from COVID(+) patients (Supplementary Fig. 6a, c). Hierarchical clustering suggested a tight association between *HIF-1α* and anaerobic glycolysis (Supplementary Fig. 6b), indicating that oxygen-deprived condition in the BALF environment is linked to anaerobic glucose metabolism. GSEA analysis showed that effector CD8+Tc were comparatively less dependent on mitochondrial metabolism during SARS-CoV-2 infection (Supplementary Fig. 6c). Reduction of NAD+ to NADH conversion is required to preserve cellular redox homeostasis and sustain glycolytic flux[39]. We observed decreased expression of transcripts encoding NADH oxidoreductases (*NDUFB8, NDUFC2,* and *NDUFA11*) in effector CD8+Tc from COVID(+) patients (Supplementary Fig. 6a, c). There was also downregulation of lipid metabolism-associated genes (*FABP4, APOC1, APOE, MARCO*) in COVID-19 effector CD8+Tcs (Supplementary Fig. 6a, c). Increased oxidative stress was also evident by overexpression of *NFE2L2* and *PRDX2* (Supplementary Fig. 6d). Decreased NADH

oxidation and a concomitant increased NAD+ are associated with impaired cytokine secretion, cell proliferation, and exhaustion[47,48]. Indeed, the expression of *CD38*, an NAD+ hydrolase linked to Tc exhaustion[48], was increased in COVID-19 effector CD8+Tc (Supplementary Fig. 6a, b). These results prove that hypoxia-induced CD38 expression is associated with metabolic reprograming and cellular exhaustion in the lung of COVID(+) patients. This conclusion is supported by higher levels of exhaustion markers *LAG3* and *TIGIT* in effector CD8+Tc from severe COVID(+) patients (Supplementary Fig. 6a, b). In order to validate the observed transcriptomic changes in the BALF, we reanalyzed an existing single-cell transcriptomic profiling dataset from COVID-19 patient PBMCs[49] (see Methods for details). In bulk CD8+Tc identified by unsupervised clustering, we also observed a strong increase in the expression of key glycolytic genes in COVID(+) patients (Supplementary Fig. 6e, f). Additionally, the expression of transcripts regulating cellular exhaustion (TIGIT, BTL4, PDCD1, and HAVCR2) was found to be similarly upregulated in severe COVID-19 (Supplementary Fig. 6g). Altogether, these results consistently suggest that hypoxia arising from COVID-19- pulmonary dysfunction augments glycolytic flux, impairs FAO and oxidative stress, leading to mitochondrial dysfunction and immunological exhaustion.

**Impaired memory cell differentiation in CD8+Tc from patients with severe COVID-19.** Memory CD8+Tc (CD8+T_M) are capable of providing protective immunity against secondary viral infection[50,51]. To understand the dynamics of memory differentiation in CD8+Tc during SARS-CoV-2 infection, we performed trajectory inference and pseudo-temporal modeling analysis for BALF CD8+Tc (Fig. 4a). Differential analysis demonstrated a strong reduction of pseudotime for CD8+T_M in severe compared to moderate COVID-19 and healthy control patients (Fig. 4a–c). In contrast to reduced CD8+T_M, the enrichment of proliferating and effector CD8+Tc was identified

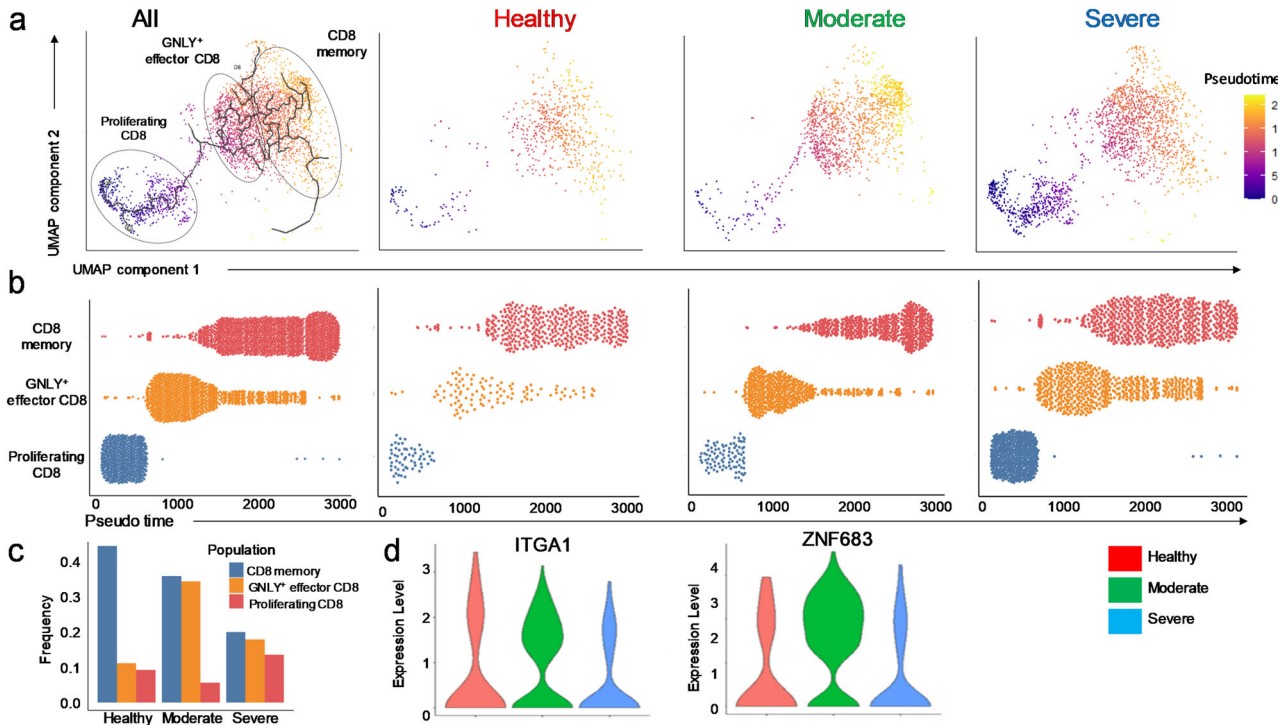

**Fig. 4 Impaired memory differentiation of CD8+Tc in COVID-19.** Pseudotime and trajectory inference analysis to evaluate the kinetic differentiation of CD8+Tc in the BALF during SARS-CoV-2 infection. **a**, **b** UMAP projection (**a**) and dot plot showing pseudotime values (**b**) of 3694 CD8+Tc of reintegrated, healthy, moderate, or severe COVID-19 patients. **c** Bar graphs displaying the frequency of CD8+Tc subpopulations across disease conditions; **d** Violin plot demonstrating the expression of tissue-resident memory encoding genes in CD8+$T_M$.

in moderate and severe compared to healthy BALF cells (Fig. 4c). Notably, these findings are also corroborated in patient PBMCs, where the proportion of effector CD8+Tc are increased in severe and moderate patients compared to healthy donors, however, the proportion of memory cells are severely decreased in comparison to both COVID(−) and healthy patients (Fig. 1e–g). These findings suggest that CD8+Tc are stalled along their memory differentiation trajectory and are unable to reach the terminal state during severe COVID-19 infection. During viral infection, circulating memory cells migrate to the infected tissue and differentiate into tissue-resident memory ($T_{RM}$) cells to provide the first response against pathogen reencounter[52]. We observed reduced expression of tissue residence- indicating genes *ITGA1 and ZNF683* selectively in severe COVID(+) patients (Fig. 4d). This result suggests that impaired differentiation of $T_{RM}$ was also evident in the lung of severe COVID-19 patients, which may be implicated with attenuated capacity for viral clearance of CD8+Tc during SARS-COV-2 infection. However, no dysregulation of memory differentiation was found in CD4+Tc of COVID(+) patients (Supplementary Fig. 7a, b), suggesting this phenomenon selectively occurred in CD8+Tc.

As memory differentiation is strictly regulated by metabolism[53,54], we investigated the potential relationship between infection-induced metabolic dysregulation and memory cell differentiation in CD8+Tc. Increased glucose uptake, indicated by augmented glut-1 expression, as well as elevated dependence on glucose metabolism, demonstrated through single-cell SCENITH analysis, were observed in CD8+$T_M$ from COVID(+) patients (Fig. 5a, b). Additionally, a decreased capacity of mitochondria to oxidize amino acid (AA) and FA were found in COVID-19+ CD8+$T_M$ (Fig. 5b). During cellular stress, excessive ROS production may be associated with increased electron leakage at the sites of complex I (NADH-ubiquinone oxidoreductase) and complex III (ubiquinone-

cytochrome c oxidoreductase) in the electron transport chain (ETC), resulting in dysfunctional mitochondrial activity[55,56]. Our results demonstrated increased ROS was solely observed in CD8+$T_M$ from COVID(−) but not from COVID(+) patients (Fig. 5c) suggesting that ROS accumulation in CD8+$T_M$ is nonspecific for COVID-19. However, a selective increase in COVID(+) patients of voltage-dependent anion channel (VDAC), involved in cellular redox and mitochondria-mediated apoptotic signaling, was identified (Fig. 5d), suggesting impaired mitochondrial integrity in COVID-19(+)CD8+$T_M$. Accordingly, decreased mitochondrial membrane potential was identified, further validating hampered mitochondrial function in CD8+$T_M$ during SARS-CoV-2 infection (Fig. 5e). Evaluating the impact of dysregulated metabolism on cellular function, we found CD8+$T_M$ exhibited increased cellular exhaustion during SARS-CoV-2 infection, as evidenced by upregulated expression of lag-3 (Fig. 6a). Consistently, CD8+$T_M$ from COVID(+) patients demonstrated impaired cytolytic function, illustrated by decreased proportion of granzyme B (GrzmB)+ cells (Fig. 6b). Mitochondrial mass is a key regulator of cytokine-secreting capacity of CD8+$T_M$[57]. We indeed observed an elevated IFNy secretion in COVID(+) CD8+$T_M$ with preserved mitochondrial mass (Fig. 6c). SCENITH analysis further confirmed that lag-3^high^CD8+$T_M$ were more metabolically dependent on glucose metabolism in COVID(+) patients (Fig. 6d). Meanwhile, oxidation of FAO and AA in mitochondria was significantly reduced in lag-3^high^CD8+$T_M$ (Fig. 6d), validating that a prolonged shift towards glucose metabolism is a key driver of mitochondrial impairment in exhausted CD8+$T_M$ during SARS-CoV-2 infection.

**Metabolic dysregulation triggers CD8+$T_M$ exhaustion in COVID-19.** GSEA analysis revealed that CD8+$T_M$ from the BALF of severe or moderate COVID(+) patients were highly dependent

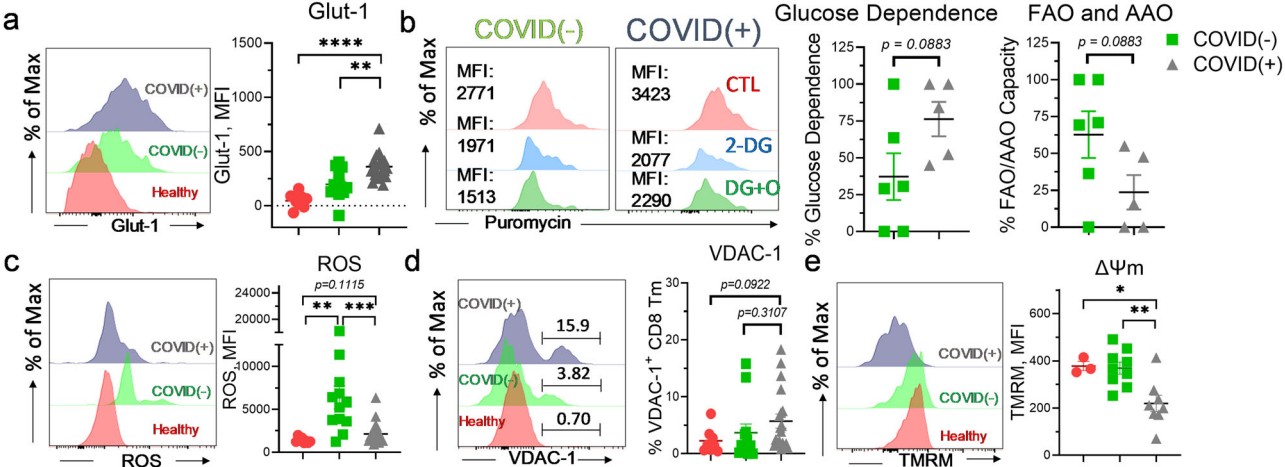

**Fig. 5 Metabolic reprogramming towards anaerobic glycolysis upon mitochondrial dysfunction in CD8$^+$T$_M$ during SARS-CoV-2 infection.** Freshly isolated PBMCs were evaluated for metabolic analysis using flow cytometry. **a** Representative histogram and summary graphs demonstrating glut-1 expression in CD8$^+$T$_M$ from healthy, COVID(−), and COVID(+) patients ($n = 8$ HD, 12 C−, 19 C+). **b** Representative histograms and dot plot graphs demonstrating glucose dependence and FAO/AAO capacities of CD8$^+$T$_M$ from healthy, COVID(−), and COVID(+) patients by the SCENITH assay ($n = 6$ C−, 5 C+). **c–e** Representative histogram and summary graphs demonstrating the expression of ROS ($n = 8$ HD, 12 C−, 19 C+) (**c**), abundance of vdac$^+$ CD8$^+$Tc ($n = 8$ HD, 12 C−, 19 C+) (**d**), and mitochondrial membrane potential ($n = 3$ HD, 9 C−, 8 C+) **e** in CD8$^+$T$_M$ from healthy, COVID(−), and COVID(+) patients. **b**, **e** three samples were excluded due to critically low CD8$^+$T$_M$ count; in (**a**, **c**, **d**), one sample was excluded due to critically low CD8$^+$T$_M$ count. Two-tailed student's $t$ test was used. *$p < 0.05$, **$p < 0.01$, and ***$p < 0.001$.

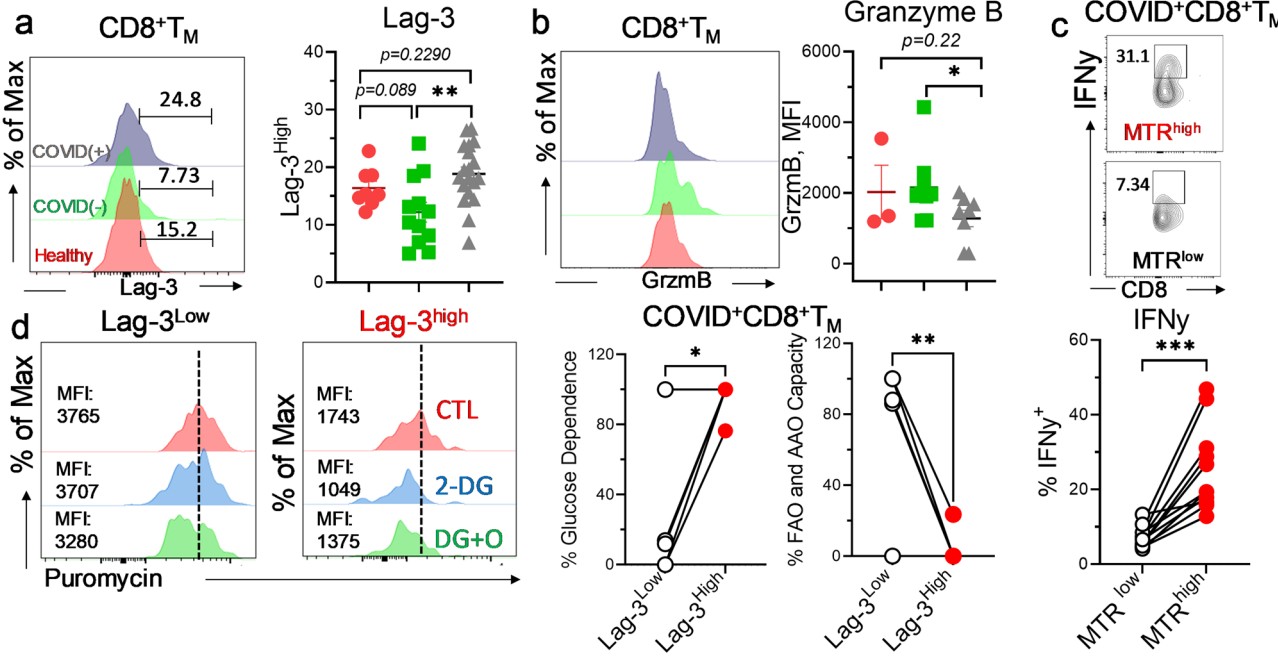

**Fig. 6 Metabolic dysregulation triggers functional impairment in CD8$^+$T$_M$.** Freshly isolated PBMCs were evaluated for cellular function and metabolic phenotype by flow cytometry. **a**, **b** Representative histogram and summary graphs demonstrating the expression abundance of lag-3$^+$ CD8$^+$T$_M$ ($n = 8$ HD, 12 C−, 19 C+) (**a**) and grzmB expression (n = 3 HD, 9 C−, 8 C+) (**b**) in CD8$^+$T$_M$ from healthy, COVID(−), and COVID(+) patients. **c** Representative contour plot and summary paired graph demonstrating IFNy$^+$ cells in MTR$^{low}$ and MTR$^{high}$ CD8$^+$Tc from COVID(+) patients ($n = 11$). **d** Summary paired graphs demonstrating glucose dependence and FAO/AAO capacities of lag-3$^{low}$ and lag-3$^{high}$ CD8$^+$T$_M$ from COVID(+) patients ($n = 5$). **b**, **d** three samples are excluded due to critically low CD8$^+$T$_M$ count; in **a**, one sample was excluded due to critically low CD8$^+$T$_M$ count. Two-tailed student's $t$ test was used. *$p < 0.05$, **$p < 0.01$, and ***$p < 0.001$.

on glycolysis for their bioenergetics demands (Fig. 7a, b). Clustering and shared upregulation of glycolytic enzyme encoding genes *GALM, GAPDH, GPI, and ALDOA* together with *HIF1A*, and transcripts regulating exhaustion (*TIGIT* and *LAG3*) (Fig. 7b) suggested that hypoxia/anaerobic axis is associated with impaired CD8$^+$T$_M$ function in COVID-19 BALF. FAO and OXPHOS

promote the development of CD8$^+$T$_M$ after antigen exposure[58]. Indeed, genes-encoding regulators of lipid uptake (*APOE*, and *APOC1*) and FAO (*OLR, MARCO, FABP4*) were downregulated in CD8$^+$T$_M$ from severe COVID(+) patients (Fig. 7a, c). Consistently, decreased expression of OXPHOS-coding genes was found in severe COVID (+) CD8$^+$T$_M$ (Fig. 7a, c). A negative

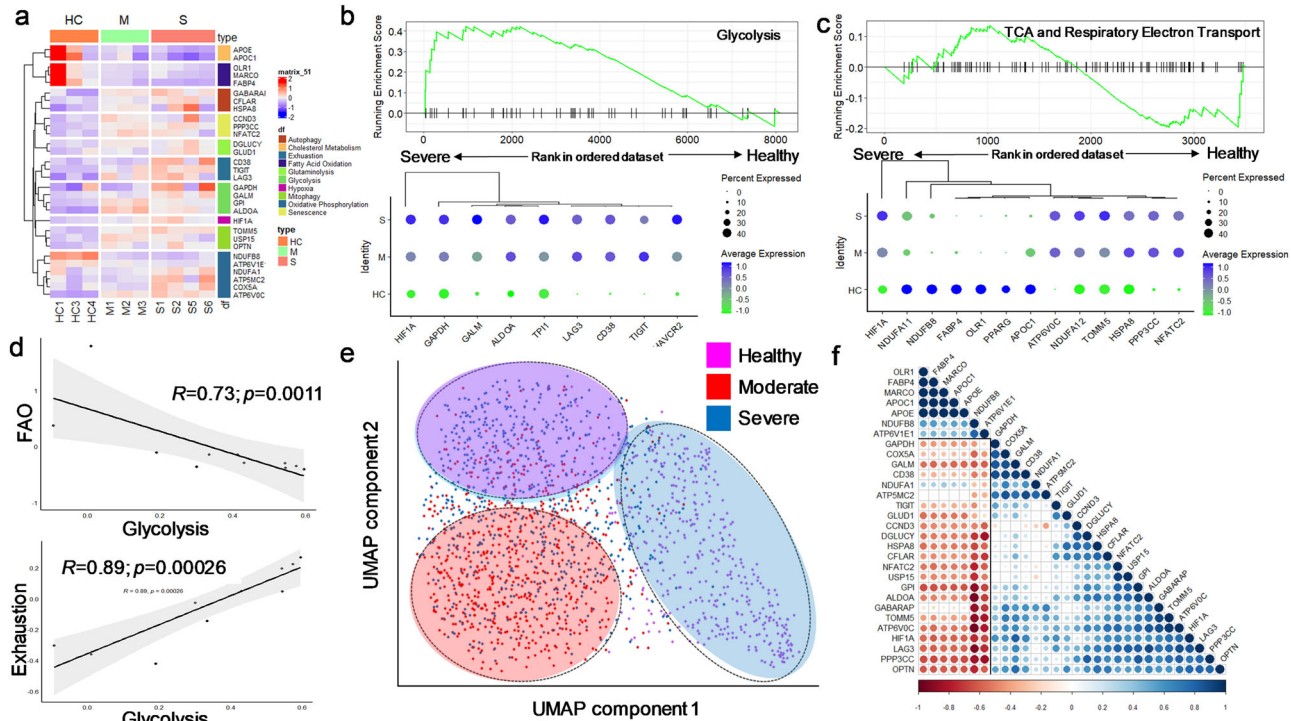

**Fig. 7 Metabolic reprogramming in BALF CD8+T_M during SARS-CoV-2 infection. a** Heatmap displaying the expression of key metabolic genes of CD8+T_M. **b–c** GSEA enrichment and hierarchical clustering plot for glycolysis (**b**) and TCA and Respiratory Electron Transport (**c**) to compare severe COVID-19 vs. healthy patients. **d** Linear regression and Pearson correlation analysis between module scores for glycolysis and exhaustion or FAO. **e** UMAP projection of CD8+T_M clustered on 42 differentially expressed metabolic genes, circles were manually drawn to highlight clustering **f**. Pearson matrix showing a correlation between differentially expressed genes.

correlation of FAO-coding transcripts and *HIF1A* expression (Fig. 7c) indicated that impaired mitochondrial metabolism is a result of hypoxia during SARS-CoV-2 infection. Strikingly, Pearson analysis revealed a negative correlation ($R = -0.73$, $p = 0.011$) between module scores for glycolysis and FAO (Fig. 7d), which further emphasizes a potential association between prolonged anaerobic glycolysis and reduced mitochondrial fitness. In support of this, a strong positive correlation between module scores for glycolysis and exhaustion ($R = 0.85$, $p = 0.00026$) (Fig. 7d) validates that excessive glycolytic dependence is tightly associated with CD8+T_M exhaustion. Genes involved in cellular senescence and mitophagy were upregulated in severe COVID-19 CD8+T_M (Fig. 7a), implying that CD8+T_M metabolically switch to these pathways in response to impaired mitochondrial metabolism. Likewise, glutaminolysis was used as an alternative pathway, evidenced by upregulation of glutamate oxidation-regulating genes (*GLUD1*, *DGLUCY*) in COVID-19(+) CD8+T_M (Fig. 7a).

PCA analysis performed on 30 differentially expressed metabolic genes (Supplementary Table 3) showed distinct clustering of CD8+T_M across different groups, further highlighting metabolic disorder during SARS-CoV-2 infection (Fig. 7e). Pearson correlation analysis showed a positive correlation between expression of genes- regulating glycolysis, mitophagy, senescence, and glutaminolysis (Fig. 7f). These genes were inversely correlated with transcripts regulating FAO and NADH oxidation (Fig. 7f). Increased CD38 expression in severe COVID-19 CD8+T_M was closely clustered with exhaustion-coding genes (LAG-3, TIGIT) (Fig. 7b) suggesting CD38 expression is associated with metabolic reprograming, memory impairment, and cellular exhaustion of CD8+T_M in the lung of COVID(+) patients. Together, these data demonstrated that the hypoxia/anaerobic glycolysis axis mediates CD8+T_M

cellular dysfunction and exhaustion at transcriptomic level in COVID-19.

**Pharmacological inhibition of mitophagy enhances cellular function of SARS-CoV-2-specific- CD8+Tc.** SARS-CoV-2 antigen-specific CD8+Tc in acute patients governs the intensity of adaptive immune response against SARS-CoV-2 infection[59]. We thus activated COVID(+) PBMCs with a CD8a SARS-CoV-2 spike peptide megapool[60] (Methodology). We found that SARS-CoV-2 CD8a peptide induced glut-1 expression in CD8+Tc from severe COVID(+) patients, suggesting that glucose metabolism is associated with SARS-CoV-2-specific CD8+Tc during infection (Fig. 8a). Given increased mitophagy has been found in CD8+Tc from PBMCs (Fig. 2g) and BALFs (Supplementary Fig. 6a, c) of COVID-19 patients, we investigated whether treatment with mitophagy inhibitor mdivi-1 could enhance the cellular function of CD8+Tc and CD8+T_M after activation with SARS-CoV-2 peptide megapool. Indeed, we found that mdivi-1 improved the generation of SARS-CoV-2-specific CD137+ cells (Fig. 8b), induced proliferation (Fig. 8c), and increased IFNγ- secreting capacity CD8+Tc (Fig. 8d). Interestingly, mdivi-1 treatment augmented CD8+T_M proliferation (Fig. 8e). Glut-1 expression was reduced in CD8+T_M under mdivi-1 treatment (Fig. 8f), suggesting that inhibition of mitophagy may reverse dysregulated glucose metabolism and normalize memory differentiation in CD8+Tc. Collectively, these results demonstrate that pharmacological inhibition of mitophagy may restore metabolic dysregulation to improve the efficacy of the CD8+Tc response in COVID-19.

**Aberrant metabolism causes NKT dysfunction in COVID-19.** NKTs, expressing CD56 and CD8, are intermediate between the

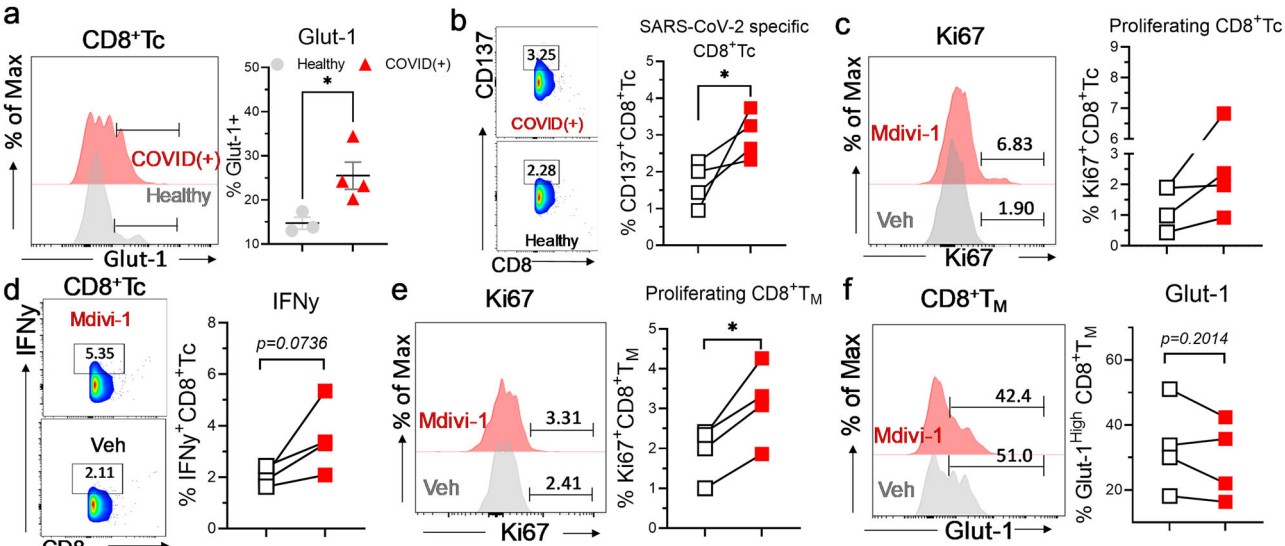

**Fig. 8 Mitophagy inhibition restores CD8⁺Tc and SARS-CoV-2-specific CD8⁺Tc function.** PBMCs from COVID-19(+) patients were activated with SARS-CoV-2 megapool CD8a peptide (see Methods) in the presence or absence of mdivi-1 (20 µM). One sample was excluded from analysis due to critically low CD8⁺Tc count. **a** Representative histogram and summary graphs demonstrating the expression of glut-1 in CD8⁺Tc from healthy and COVID(+) patients (n = 3 HD, 4 C). **b–d** Representative histogram/density plots and graphs demonstrating the abundance of antigen-specific (CD137⁺) (n = 4 C) (**b**), Ki-67⁺ (n = 4 C) (**c**), and IFNy⁺ (n = 4 C) (**d**) cells among CD8⁺Tc of vehicle or mdivi-1 vs vehicle-treated samples. **e, f** Representative histogram and summary graphs demonstrating the abundance of Ki-67⁺ (n = 4 C) (**e**) and glut-1⁺ (n = 4 C) (**f**) cells amongst CD8⁺T_M of vehicle or mdivi-1-treated samples. Two-tailed student's t test was used. *p < 0.05, **p < 0.01, and ***p < 0.001.

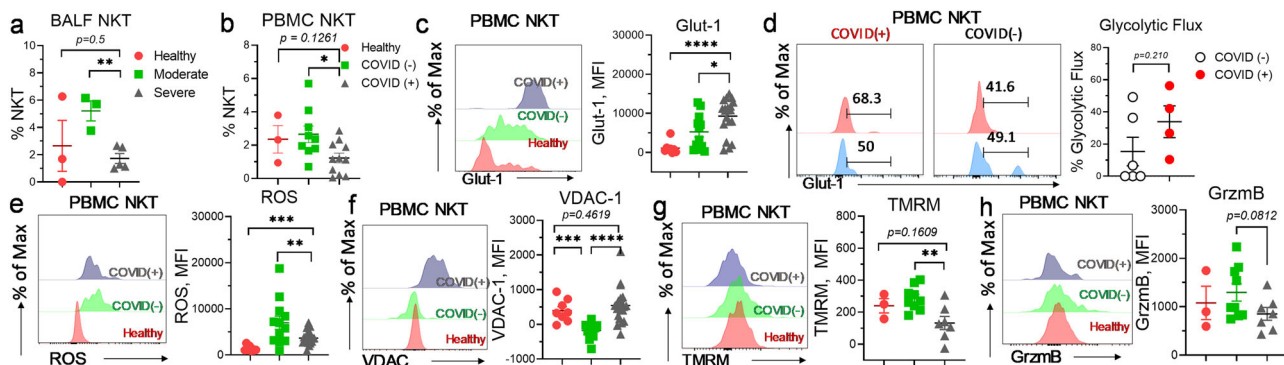

**Fig. 9 Circulating NKTs exhibit dysfunctional immunometabolic phenotype in COVID-19. a** Summary graph demonstrating the frequency of NKT in PBMCs from healthy, COVID(−), and COVID(+) patients (n = 3 HD, 3 M, 4 S). **b** Summary graph demonstrating the frequency of NKT in the BALF from healthy, moderate, and severe patients (n = 3 HD, 9 C−, 11 C+). **c** Summary graphs and histograms demonstrating the expression of glut-1 in NKTs from healthy, COVID(−), and COVID(+) patients (n = 8 HD, 12 C−, 20 C+). **d** Histograms and graphs demonstrating glycolytic flux of NKTs from COVID(−) and COVID(+) patients from the SCENITH assay (n = 6 C−, 4 C+). **e–h** Histograms and graphs demonstrating the expression of ROS (n = 8 HD, 12 C−, 20 C+) (**e**), VDAC (n = 8 HD, 12 C−, 20 C+) (**f**) TMRM (n = 3 HD, 9 C−, 7 C+) (**g**), and grzmB TMRM (n = 3 HD, 9 C−, 7 C+) (**h**) in NKTs from healthy, COVID(−), and COVID(+) patients. **d**, **f**, **g**, **h** Four samples were removed from analysis due to a critically low NKT cell count. Two-tailed student's t test was used. *p < 0.05, **p < 0.01, and ***p < 0.001.

CD8 and NK cell lineages[19]. NKTs play critical roles in preventing pneumonia during chronic pulmonary disease[61]. SARS-CoV-2 infection impairs NKTs effector functions and hinders the effective clearance of virally infected cells[19]. Circulating NKT frequency has been implicated as a powerful prognostic biomarker of COVID-19 severity[18]. Consistently, in our current cohort, NKT lymphopenia was observed in both PBMCs and BALF from COVID(+) patients (Fig. 9a, b). Examination of the metabolic profile of NKTs from patient PBMCs revealed a significant increase in the expression of glut-1 in NKTs from COVID(+) patients compared to those from both healthy and COVID(−) counterpart (Fig. 9c), suggesting that augmented glucose utilization by NKTs has selectively occurred during SARS-CoV-2 infection. SCENITH analysis validated a trending

increase in the glycolytic flux of COVID(+) NKTs (Fig. 9d). Consistently, enhanced ROS accumulation was accompanied by a concurrent upregulation of VDAC, indicative of impaired mitochondrial fitness was detected in COVID(+) NKTs (Fig. 9e, f). Accordingly, mitochondrial membrane potential was found reduced in NKT from COVID-19 patients (Fig. 9g). NKTs exhibit protective activity against viral infection via the secretion of cytolytic molecules such as grzmB[62]. Consequently, decreased expression of grzmB in COVID(+) NKTs indicated reduced effector function of this cell subset during SARS-CoV-2 infection (Fig. 9h). To further probe the perturbed metabolism in NKTs from COVID-19 patients, we characterized gene expression profiles of BALF NKTs. Co-expression of *CD8A and KLRD1* was used to define NKT lineage (Supplementary Table 4). A

transcriptional program associated with hypoxia-induced metabolic reprogramming was seen in NKTs from severe COVID-19 patients (Supplementary Fig. 8a). Consistent with increased glut-1 expression (Fig. 9c), upregulation of anaerobic glycolytic metabolism involving genes (LDHA, TPi, PGAM1, ALDOA) were identified in severe COVID-19 NKTs (Supplementary Fig. 8a). GSEA analysis validated a strong, concomitant increase in glycolysis genes (Supplementary Fig. 8b). Importantly, a substantial increase in the expression of genes- regulating TCA and respiratory electron transport was also seen (Supplementary Fig. 8b). Under normal conditions, activated NKTs will expectedly use pyruvate dehydrogenase (PDHA1/2) to supply acetyl-CoA for mitochondrial TCA cycle metabolism[63]. However, under reduced oxygen saturation in the lung of COVID-19 patients, downregulation of genes encoding lipid uptake, FAO, and NADH oxidation (NDUFB8, NDUFA11, NDUFA13) was found in severe COVID(+) NKTs (Supplementary Fig. 8b,c) suggesting mitochondrial dysfunction in NKTs during SARS-CoV-2 infection. This observation may illustrate that glycolysis-derived pyruvate is converted into lactate rather than oxidized in TCA cycle, leading to a lack of materials for sufficient OXPHOS to support NKT effector function during SARS-CoV-2 infection. Alternatively, increased levels of GADD45B and SLC25A5 transcripts (Supplementary Fig. 8a) suggest metabolic adaptation via enhancement of mitophagic activity in NKTs during SARS-CoV-2 infection.

**Metabolic dysregulation impairs immune surveillance and increases pro-inflammatory response in lung epithelial cells during SARS-CoV-2 infection**. Epithelial cells (ECs) secrete cytokines and help mediate antigen presentation to modulate immune cells' function during viral infection[64]. Differential expression analysis revealed overexpression of key immune signaling pathways in COVID-19 ECs (Supplementary Fig. 9a). Induction of a pro-inflammatory cascade including type 1 IFN, toll-like receptor, NF-kB, and chemokine signaling was observed in COVID-19 ECs (Supplementary Fig. 9b). Glucose metabolism mediates type I IFN secretion through enhancing NF-kB expression[65] and epigenetic acetylation[66]. Indeed, we found a positive correlation between module scores for glycolysis and type 1 IFN signaling (Supplementary Fig. 9c) as well as for glycolysis and NF-kB signaling (Supplementary Fig. 9d). Chronic presentation of viral antigens to CD8+Tc by ECs may cause cellular dysfunction[67]. We observed that genes encoding HLA class 1 (HLA-E, PSMA-6, TAP1, IFI30) were enriched in COVID-19 ECs (Supplementary Fig. 9a). GSEA analysis further confirmed the upregulation of HLA class 1 antigen presentation in bulk ECs (Supplementary Fig. 9e). In contrast, downregulation of genes encoding HLA class 2 (HLA-DRA, HLA-DPA1, HLA-DMA, DYNLL1) was found in COVID-19 ECs (Supplementary Fig. 9a) which was further confirmed by GSEA analysis (Supplementary Fig. 9e). Glycolysis was reported to repress functional response of antigen-presenting cells during infection[68]. We indeed observed a negative correlation between glycolysis and genes encoding HLA class 2 machinery (Supplementary Fig. 9f). These results revealed potential links between dysregulated EC metabolism with cytokine release syndrome and immune dysfunction in COVID-19. Network analysis demonstrated a connection of SARS-CoV-2 infection with attenuated transcriptional factor network demonstrated by downregulation of the transcriptional factors ZKSCAN1 and CSNK2B, and upregulation of KLF6, NEAT1, and JUND (Supplementary Fig. 9g).

BALF ECs were next identified and subsetted for downstream analysis (Supplementary Fig. 10a–c). Differential expression analysis revealed key differences in the expression of transcripts governing key metabolic pathways (Supplementary Fig. 10d).

Additionally, UMAP performed solely on differentially expressed metabolic genes revealed distinct clustering of bulk epithelial cells along disease severity (Supplementary Fig. 10e). Pearson correlation analysis performed on ECs revealed a strong positive correlation between HIF1A and key glycolytic transcripts, suggesting a hypoxia- induced glycolytic metabolic reprogramming (Supplementary Fig. 10f). ECs were then divided into pseudostratified ciliated and nonciliated subtypes based on the expression of canonical genes associated with cilia production (CFAP126, and DNAAF) (Supplementary Fig. 10b). The ratio of pseudostratified ciliated ECs to nonciliated epithelial cells was inversely correlated with COVID-19 disease severity (Supplementary Fig. 10c). This finding suggested that SARS-CoV-2 infection produced direct injury to the ciliated EC compartment. Overexpression of glycolytic transcripts (ENO1, ADH1A3, GAPDH, ALDOA, PCK2) was noted in both ciliated and nonciliated EC subsets from COVID(+) patients (Supplementary Fig. 10g, i). These results were validated by GSEA analysis, which demonstrated enrichment of glycolysis genes (Supplementary Fig. 10g, i). We also observed decreased expression of FAO regulating genes to different extents in ciliated and nonciliated ECs from severe COVID(+) compared to healthy control (Supplementary Fig. 10h, j). HIF-1A and anaerobic glycolysis gene expression was strongly correlated with reduced expression of the OXPHOS and TCA cycle genes in these EC subsets from severe COVID-19 (Supplementary Fig. 10h, j). GSEA analysis demonstrated enrichment of glycolysis, as well as a large downregulation of OXPHOS and TCA cycle regulating genes in ciliated and nonciliated ECs of severe COVID-19 patients (Supplementary Fig. 10h, j). Collectively, these results suggested that oxygen-deprived conditions in the COVID-19 lung mediates a metabolic switch from aerobic FAO and OXPHOS towards anaerobic glycolysis in ECs, which is strongly linked to mitochondrial dysfunction.

## Discussion

Metabolic comorbidities have been identified as key risk factors for COVID-19 severity and mortality[5,69]. However, how metabolic dysregulation in patients is linked to worsened immunopathology during SARS-CoV-2 infection remains unclear. A more comprehensive understanding of the mechanisms underlying this link would provide critical insight for the prognosis and therapeutic treatment of COVID-19. Here, using a tripartite combination of multiparametric flow cytometry, SCENITH single-cell metabolomics, and scRNA-SEQ re-analysis, we show immune dysregulation in SARS-CoV-2 infection is associated with metabolic reprogramming in CD8+Tc and NKTs, both of which have critical roles in the anti-viral adaptive immune response[18,70]. Notably, this metabolic dysfunction was absent in patients negative for COVID-19 who demonstrated COVID-19-like upper respiratory symptoms, validating that these altered immunometabolic profiles were not merely a consequence of increased inflammation, but rather uniquely specific to SARS-CoV-2 infection.

Despite CD4+Tc lymphopenia having been reported in severe COVID-19 patients in a number of studies[70,71], it is highly controversial whether this phenomenon occurs in CD8+Tc. In this study, using high-dimensional flow cytometry to validate and decipher highly resolved CD8+Tc subsets, we demonstrated clear, decreased cell frequency amongst CD8+TM and NKT subsets during SARS-CoV-2 infection. Interestingly, we observed an increase in the percentage of effector CD8+Tc in both COVID-19 patient PBMCs and severe BALFs compared to healthy individuals, indicating that the initial differentiation of effector CD8+Tc is not impaired in COVID-19. However, this is in stark

contrast to the frequency of CD8$^+$T$_M$, which was found to be heavily decreased compared to healthy and COVID(−) patients. Thus, our results suggest that dysfunction in both CD8$^+$Tc and effector CD8$^+$Tc leads to a severe impairment in CD8$^+$T$_M$ differentiation. Mechanistically, pseudo-temporal modeling and trajectory inference analysis demonstrated that CD8$^+$Tc are stalled on their differentiation trajectory towards memory cells in severe SARS-CoV-2 infection. Noticeably, CD8$^+$T$_M$ and NKT lymphopenia was not identified in COVID(−) patients, suggesting that a reduced abundance of CD8$^+$T$_M$ and NKTs are specific predictive biomarkers for COVID-19.

Hyperglycemia during hospital admission is a strong predictor of COVID-19 mortality[5,72]. Consistently, increased activity of LDH, a gate-keeping glycolytic enzyme, is associated with pulmonary dysfunction during COVID-19[73]. These evidence highlight the lung dysfunction-induced hypoxia/anaerobic glycolysis axis as a key mechanism mediating dysregulated host immunometabolism during SARS-CoV-2 infection. Noticeably, Cossarizza et al. failed to detect metabolic changes in CD8$^+$Tc using Seahorse bioenergetics analysis[74]. Additionally, using scRNA-SEQ and flow cytometry, the Powell group did not observe attenuation in glucose metabolism of CD8$^+$Tc[75]. This discrepancy can probably be attributed to the fact that Tc were stimulated with αCD3/CD28 polyclonal activation under normoxic conditions which may result in nulling of any potential metabolic differences present in COVID-19. Another potential explanation arises from the fact that cellular metabolism in these studies was assessed for entire Tc populations, which are highly heterogeneous amongst subsets with respect to their metabolism. To this end, in our current study, using SCENITH single-cell metabolomics to evaluate the bioenergetics flux of freshly isolated COVID-19$^+$ PBMCs, we successfully elucidated a systematic metabolic reprogramming characterized by excessive glucose metabolism accompanied with impaired mitochondrial fitness, resulting in subsequent cellular exhaustion specific to CD8$^+$T$_M$ and NKT cells. Our single-cell metabolomics approach allowed us to directly probe the metabolism of exhausted CD8$^+$Tc and to validate a large overdependence on glucose metabolism as a hallmark of CD8$^+$Tc exhaustion in COVID-19. Moreover, a strong increase in CD8$^+$Tc hif-1α expression in patients with metabolic syndrome, as well as mild correlations between lag-3 and VDAC-1 expression with serum glucose level, may a mechanistic justification for impaired antiviral immunity against SARS-CoV-2 infection in hyperglycemic and diabetes patients. Importantly, a metabolic shift towards increased anaerobic glycolysis occurred selectively in COVID-19 patients, compared to other COVID(−) patients with similar respiratory symptoms. Recently, Siska et al. reported that SARS-CoV-2 infection mediates increased hypoxia-induced mitochondrial ROS through the enhancement of basigin CD147 expression, resulting in mitochondrial stress and cellular dysfunction in Tc of patients with severe SARS-CoV-2[9]. In contrast, our current study demonstrated that augmented cytosolic ROS level was the most apparent in CD8$^+$Tc from COVID(−) patients, suggesting that ROS accumulation in CD8$^+$Tc is a more general feature of increased lung inflammation. Presumably, the combination of ROS accumulation with a COVID-19-specific hypoxia triggered a shift to anaerobic glycolysis, which is a primary factor driving immunometabolic dysfunction in CD8 lymphocytes during SARS-CoV-2 infection.

CD8$^+$T$_M$ are critical for long-term protection against viruses and strongly correlate with immune protection[76]. However, little is known about the SARS-CoV-2-specific Tc immunity in prior, virally exposed individuals, such as how CD8$^+$T$_M$ are generated post-acute infection. Normally, CD8$^+$T$_M$ depend on OXPHOS and oxidation of intracellular lipids in mitochondria to sustain

their energetic demands[77,78]. However, in the current study, as a consequence of reduced oxygenation due to COVID-19 lung dysfunction, both CD8$^+$T$_M$ and their upstream effector precursors were found to be phenotypically and functionally glycolytic, suggesting a metabolically linked impairment in CD8$^+$T$_M$ function and differentiation. Accordingly, CD8$^+$Tc activated by SARS-CoV-2- specific peptide showed significantly upregulated glucose uptake. Given that healthy, functional SARS-CoV-2-specific CD8$^+$Tc are associated with milder, recovered, and convalescent COVID-19 patients[79], targeting the hypoxia/anaerobic glycolysis may potentially improve the function of SARS-CoV-2 antigen-specific CD8$^+$Tc during re-exposure to viral antigens, and rescue impaired memory cell differentiation. Supporting this hypothesis, cyclophilin A was found to restore SARS-CoV-2-specific CD8$^+$Tc function through normalizing cellular metabolism[9]. Further, a recent stage-2 clinical trial using 2-DG, a competitive inhibitor of glycolytic flux, as a therapeutic treatment for COVID-19 was successful in improving patient outcomes[80].

The current study strongly implicates mitophagy as a potential therapeutic target for COVID-19 treatment. Mitophagy is the cellular process involved in selective autophagic degradation of dysfunctional mitochondria[81]. Depletion of impaired mitochondria via mitophagy redirects metabolism towards increased glucose utilization[82,83]. Accordingly, ablation of mitophagy may potentially attenuate CD8$^+$Tc exhaustion and improve memory cell differentiation. Indeed, pharmacological targeting of mitophagy by mdivi-1 restored the proliferation, activation, and memory formation of CD8$^+$Tc and CD8$^+$T$_M$, as well as enhanced the generation of SARS-CoV-2- specific CD8$^+$Tc via the attenuation of glucose metabolism. Furthermore, unlike glycolysis, mitophagy is not critical for initial Tc activation and effector cell differentiation[84]; thus, mitophagy-targeting approaches can potentially be used immediately after infection. Additionally, virus have been found to hijack intracellular mitophagy to attenuate innate immune response activity and promote viral proliferation[85]. Indeed, mdivi-1 has been shown to effectively reduce SARS-CoV-2 replication in virally infected cells[17]. Thus, therapeutic use of mdivi-1 for COVID-19 may have a dual effect of 1) improving the efficacy of the adaptive Tc immune response via metabolic restoration and 2) directly inhibiting viral replication in the host.

NKT abundance is a strong predictive biomarker for COVID-19 outcome[86]. NKTs have been proven to have a key role in the prevention of COVID-19-induced pneumonia[19]. As a key bridge between innate and adaptive immunity, it is critical to understand the function and role of NKTs during SARS-CoV-2 infection. However, little is known about how NKT metabolism during infection. Consistently, in current study, NKT lymphopenia was evident in both the BALFs and PBMCs. Furthermore, COVID-19(+) NKTs demonstrated augmented hypoxia-mediated anaerobic glycolytic activity, accompanied by elevated mitochondrial impairment. However, unlike CD8$^+$Tc, a strong increase in OXPHOS was seen in NKTS from severe COVID-19 patients. This differential metabolic response can be explained by the observations that NKT cells are considerably more dependent on mitochondrial metabolism after activation in comparison to conventional Tc[21,22]. Whereas conventional Tc exhibit a tremendous Warburg-like upregulation of aerobic glycolysis upon activation[87], NKTs remain heavily reliant on mitochondrial respiration[21,22]. Thus, upon the onset of systemic hypoxia after initial lymphocyte activation, OXPHOS-dependent NKTs are likely unable to sustain mitochondrial activity and thus also upregulate glycolysis to sustain their bioenergetics demands. Our results thus highlight that this metabolic reprogramming is associated with a decrease in the frequency of NKTs, as well an impairment in cytolytic function.

Interestingly, we found that SARA-CoV-2 derived EC damage creates oxygen-deprived conditions in the lungs that not only induce metabolic reprogramming of various immune cell subsets, but also themselves. We found that during COVID-19 infection, differential metabolism drives lung ECs towards senescence and towards acquiring a SASP phenotype, leading to secretion of pro-inflammatory cytokines, reduced HLA class 2 mediated immunosurveillance, and increased HLA class 1 machinery. Prolonged stimulation of exhausted lymphocytes, which demonstrate attenuated effector function and cytokine secretion in nutrient-depleted microenvironments, by antigen presenting cells via HLA class 1 leads to increased cellular exhaustion[88], which further impairs the capacity of cells to differentiate into memory phenotypes. Our results, therefore, show that the immunometabolic rewiring of ECs in the BALFs can be a potential mechanism for organ-specific lymphocyte exhaustion and memory cell dysfunction. Further, this observation thus highlights that unconventional antigen presentation on non-hematopoietic ECs via HLA class 1, in addition to conventional antigen presentation by professional APCs (monocyte, DC, and macrophage), can be considered as a potential target for therapeutic development.

In summary, in the current study, we show that CD8+Tc, NKTs, and ECs undergo a global metabolic reprogramming towards anaerobic metabolic processes including glycolysis, mitophagy, and glutaminolysis. As a result, specific CD8+Tc subsets and NKTs demonstrate profound metabolically- linked exhaustion and effector function, as well as impaired differentiation into memory cells. We further validate mitophagy as potential target for therapeutic treatment of severe SARS-CoV-2. Our current study therefore sheds important light on the key molecular and cellular mechanisms by which immunometabolism regulates pathobiology in SARS-CoV-2 infection and validate the concept of targeting immunometabolism to treat acute COVID-19 severity or enhance the efficacy of COVID-19 vaccination therapies.

Shortcomings of our study included a small sample size leading to weak statistical power in analyses, as well as an inability to stratify for disease severity in the analysis. In addition, given the limited amount of blood received for patient per analysis and the high degree of lymphopenia amongst the samples, it was not possible to perform every experiment with all the patients in our cohort. We also report a limited sample size for analysis of BALF transcriptomic data, along with a lack of proteomic data for our study of lung ECs.

## Methods

**Sample acquisition.** Blood from healthy donors was ordered from Research Blood Company. Blood samples from hospitalized COVID-19 patients or patients with COVID-19-like symptoms, however, testing negative for COVID-19 (COVID(−) patients), were collected at AdventHealth Hospital under protocols IRB# 1668907 and #1590483 approved by AdventHealth IRB committee. Informed consent was taken for all patients, in addition to maintenance of strict confidentiality in accordance with HIPAA confidentiality. COVID-19 positivity/negativity was confirmed by a PCR test at AdventHealth. Blood was used for human PBMC, plasma, and serum isolation.

**Patient classification criteria.** Our cohort consisted of hospitalized COVID(+) and COVID(−) patients requiring either emergency admission or acute IP care. The classifier "severe respiratory impairment" was used to describe patients recorded as having either "dependence on respirator (ventilator)", "Supplementary oxygen", "acute respiratory failure", "acute respiratory distress syndrome", "hypoxia", "hypoxemia", "acute and chronic respiratory failure", "chronic respiratory failure". Additionally, the classifier "presence of dysfunctional lung symptoms" was used to describe patients who are recorded as having any of the above described symptoms of "severe respiratory impairment" as well as "pneumonia", chronic obstructive pulmonary disease", "pneumonitis", "pulmonary fibrosis", "bronchiectasis", "acute pulmonary edema", "interstitial pulmonary disease", "chronic pulmonary edema", "dependence on other enabling machines and devices". Patients considered as having metabolic syndrome/disorder were diagnosed as having either "obesity", "morbid obesity", "type 2 diabetes mellitus",

"other unspecified diabetes mellitus", "prediabetes", "diabetes insipidus", "type 1 diabetes mellitus", "metabolic syndrome". Patients considered as having prior transplantation and or/immunosuppression/dysfunction were diagnosed as having either "organ transplant status/failure", "bone marrow transplant", "stem cell transplant", "disorder involving the immune mechanism", "immunodeficiency", "history of immunosuppression therapy", "human immunodeficiency virus (HIV) disease".

**PBMC isolation.** PBMCs were isolated by density-gradient centrifugation using Ficoll-Paque™. Briefly, blood specimens were centrifuged at $700 \times g$ for 7 min at room temperature (RT) for serum collection. The pellets were resuspended in phosphate buffer saline (PBS). Cell suspensions were carefully overlaid on the top of 4 mL of Ficoll in 15 mL conical tube, followed by centrifugation at $700 \times g$ at RT for 25 min without break. PBMCs were collected from interphase between plasma and Ficoll layers. Cells were then washed twice with PBS to remove any Ficoll residue. All procedures were approved at the BSL2+ level by University of Central Florida Environmental Health and Safety.

**Antibody staining and flow cytometry.** The concentrations of all antibodies/dyes in flow cytometric staining were used in accordance to the manufacturers' recommendations (Supplementary Table 5). PBMCs ($0.5 \times 10^6$ cells) were first stained with live/dead in PBS for 15 min, washed with Flow Cytometry Staining Buffer (FACS Buffer), and stained with surface markers in ice-cold FACS buffer at 4 °C for 30 min. PBMCs were then washed twice with FACS buffer and stained with secondary antibodies for 15 min. Samples were fixed and permeabilized using Fixation/Permeabilization buffer (20 min) at room temperature and washed with ice-cold FACS buffer. PBMCs were then stained with intracellular antibodies at 37 °C for 45 min in permeabilization buffer. Samples were washed once with permeabilization buffer before being resuspended in FACS buffer for flow cytometric analysis using Cytoflex system. Data were then analyzed by Flowjo™v10.

**Single-cell metabolism uptake assay.** Mitochondrial membrane potential, protein translation, mitochondrial mass, and cytosolic ROS were assessed via uptake of TMRM, puromycin; mitotracker (MTR); and 2′,7′-dichlorofluorescein diacetate (DCFDA). Cells were incubated with fluorescent dye for 15 min at 37 °C before being washed with ice-cold FACS buffer followed by downstream flow cytometric staining in accordance to aforementioned description.

**Single-cell metabolomics assay (SCENITH).** PBMCs were incubated for 2 h followed by a 20 min treatment with either 100 mM 2-deoxy-glucose (2-DG), 1 μM oligomycin (O), or a sequential combination of both drugs 37 °C, 5% $CO_2$. Subsequently, puromycin (10 μg/mL) was added to the culture for additional 30 min. Afterwards, cells were washed with ice-cold PBS and subjected to downstream flow cytometric staining. Cells were fixed, permeabilized, and stained intracellularly with the monoclonal anti-puromycin antibody for 45 min. Fluorescence was recorded in the FITC channel.

Values for "glucose dependence" and "FAO and AAO capacity" were calculated in accordance with the original SCENITH protocol[89]. Briefly, "glucose dependence" was calculated as 100*($CTL^{PuroMFI}$ - 2-$DG^{PuroMFI}$)/($CTL^{PuroMFI}$ - 2-DG + $O^{PuroMFI}$). "FAO and AAO capacity" was calculated as 100 - 100*($CTL^{PuroMFI}$ - 2-$DG^{PuroMFI}$)/($CTL^{PuroMFI}$ - 2-DG + $O^{PuroMFI}$). Additionally, for direct investigation of the dependence of overall cellular energy production on glycolysis, we calculated values for "glycolytic flux" as the percent decrease in puromycin$^{high}$ cells after treatment with 2-DG compared to control as reported previously by Hong et al.[90]. The formula used for this was 100*(%Puro$^{+CTL}$ - %Puro$^{+2-DG}$)/(%Puro$^{+CTL}$). Likewise, we also calculated values for "mitochondrial flux" as percent decrease in puromycin$^{high}$ cells after treatment with oligomycin compared to control. The formula used for this was 100*(%Puro+$^{CTL}$ - %Puro+$^{Oligomycin}$)/(%Puro+$^{CTL}$).

Similar to the guidance provided by Arguello et al.[89], if the MFI value of puromycin is higher in the inhibitor treatment (either 2-DG or O) compared to the control, we considered the glucose/mitochondrial dependence value to be 0%. Additionally, if the percent of puromycin-positive cells is higher in the inhibitor treatment compared to the control, we considered the glycolytic/mitochondrial flux to be 0%. Likewise, if the MFI value of puromycin is lower in the singular inhibitor treatment (either 2-DG or O) compared to the combination treatment (2-DG + O), we considered the dependence to be 100%.

**CD3/CD28 polyclonal activation.** 96 well flat-bottom plates were coated with 5 μg/mL anti-CD3 for 2 h at 37 °C, 5% $CO_2$. PBMCs ($5 \times 10^5$) were activated with plate bound anti-CD3 and soluble anti-CD28 (5 μg/mL) in complete culture media. After 48 h, cells were stimulated with 50 ng/mM PMA and 1 μg/mL ionomycin, followed by 3 h incubation with GolgiStop to evaluate intracellular cytokine secretion. Cells were then subjected to flow cytometric staining as described above.

**SARS-CoV-2-specific peptide activation.** PBMCs ($5 \times 10^5$) were stimulated with 1 mg/mL of CD8a SARS-CoV-2 megapool peptide[91] in the presence or absence of 20 μM mdivi-1 at 37 °C, 5% $CO_2$ for 96 h.

**High-dimensional flow cytometry analysis**. First, traditional bivariate gating using FlowJo™v10 was performed to identify major cell types. CD8+Tc and NKTs from each sample were identified, isolated, and exported to new fcs files. The flowCore package in R was used to read concatenated CD8+Tc and NKT fcs files into the R environment[92]. Next, an arcsinh transformation was applied for data normalization. Data from all of the samples were then merged into one catalyst object, upon which downstream analyses were performed[93]. PCA was then run on bulk sample-aggregated data and the top three principal components were plotted. FlowSOM clustering was performed on only cell surface markers used for phenotypic identification with the number of expected populations set at 30[94]. Clusters were then annotated based on canonical marker expression. Differential abundance of cell-type proportions and differential expression of MFI values were then conducted.

**BALF scRNA-SEQ data acquisition**. Single-cell RNA-seq data from the BALF of 6 severe COVID patients, 3 moderate patients, and 4 healthy donors were used for analysis[32]. This study defined moderate and severe COVID-19 patients as those with pneumonia experiencing respiratory distress and hypoxia and with critical condition, requiring ICU care, and having been placed under mechanical ventilation, respectively. Prefiltered expression matrices with UMI counts were downloaded from the GEO Database with accession number GSE145926. Additionally, as suggested by the original study, data from an additional BALF sample derived from a healthy donor from a separate study was used as a ref. [95]. Prefiltered expression matrices with UMI counts were downloaded from the GEO Database with accession number GSE128033 and sample number GSM3660650.

**PBMC scRNA-SEQ data acquisition**. Single-cell RNA-seq data from a total of 8 COVID-19 patients and 4 healthy donors were reanalyzed from an existing study published by Lee et al.[49]. For three out of the eight COVID-19 patients, two separate samples were collected at different time points for total 11 samples (6 severe COVID-19, 5 mild COVID-19, and 4 healthy) present in the cohort for analysis. The authors defined COVID-19 disease severity as either moderate or mild using metrics from the National Early Warning Score methodology, where "respiratory rate, oxygen saturation, oxygen supplement, body temperature, systolic blood pressure, heart rate, and consciousness" were used as evaluating criteria[49,96]. Prefiltered expression matrices with UMI counts were downloaded from the GEO Database with accession number GSE149698. Metadata was downloaded from the Supplementary information provided in the original article[49].

**Quality control and preprocessing of BALF and PBMC scRNA-SEQ Data**. Quality control and data preprocessing were conducted using Seurat[97,98]. First, cells for which more than 10% of reads were mitochondrial transcripts were discarded. Next, we removed cells that had 1000 detected transcripts. Cells with less than 200 and greater than 6000 unique genes were also filtered. Filtered data from different 14 patient samples were integrated in Seurat. Individually, data from each sample was log2 normalized and the top 2000 variable genes were identified using the "vst" method in Seurat. Data from each sample was next scaled and PCA was run with percentage of mitochondrial DNA and number of detected unique genes regressed out. Alignment and batch effect correction was done using reciprocal PCA and canonical correlation analysis (CCA) (in accordance to standard Seurat integrated analysis workflow) on the first 30 dimensions of the data. Next, a shared nearest-neighbor graph was constructed and Louvain-based optimization was run to perform unsupervised clustering. UMAP was next run on the first 30 dimensions. Data was next log 2 normalized and scaled in the "RNA" assay for expression analysis, with percentage of mitochondrial DNA and number of detected unique genes regressed out. The top 2000 variable genes were determined by the "vst" method in Seurat. Expression of canonical markers were used to define cell populations. For each identified cell population, SCTransform was done on the "RNA" assay to improve normalization and aid in visualization purposes.

**Tc reintegration and secondary clustering in BALFs**. Tc were subsetted and split according to samples. Data from healthy control 1 and severe 1 were excluded from analysis due to low Tc count. To further correct for batch effect, Tc were then reintegrated using canonical correlation analysis in Seurat run on the first 30 dimensions. SCTransform was next implemented on the "RNA" assay and stored in a new "SCT" assay to better normalize counts across samples for visualization purposes with percentage of mitochondrial DNA regressed out. Standard log 2 normalization and scaling was then performed on the "RNA" assay. Subpopulations of Tc were next identified based upon canonical marker expression.

**Trajectory inference and pseudo-temporal ordering**. Monocle 3 was used to construct a trajectory upon UMAP embeddings and order cells in pseudotime[99]. Analysis was performed on both CD8+ and CD4+Tc. Seurat wrapper function "asMonocle" was used to create Monocle CellDataSet object from an existing Seurat object. "learn_graph" function was used to construct trajectory mappings onto transferred UMAP embeddings. "order_cells" was used to estimate and order cells in pseudotime. All samples for CD8+ and CD4+Tc populations were ordered together and were split by disease state after ordering for differential comparison of pseudotime.

**Metabolic phenotype-based clustering**. To investigate whether metabolic phenotypes of certain cell populations could be used as predictive indicators of disease severity, dimensionality reduction at both a single-cell and sample-wide resolution was done only on key identified differentially expressed metabolic genes to see if cells/samples would cluster according to disease severity. For sample-wide analysis, principal component analysis was conducted and the first three principal components were visualized. For analysis at single-cell resolution, UMAP was done and the first two components were visualized.

**Network analysis**. For construction of gene pathway enrichment network, networkanalyst.ca was used[100]. All statistically significant genes were inputted along with log fold change values to construct enrichment network. Transcription factor–gene interaction network was also constructed using networkanalyst.ca[100]. Statistically significant genes along with log fold changes values were inputted. The "degree" filter was first set to 100 and then the "betweenness" filter was set to 170.

**Downstream analysis**. For heatmap visualizations, scaled SCTransformed values were used and the Complexheatmap package was used to generate visualization[100]. Hierarchical clustering and dendrogram generation were performed using default settings of the package. Outliers with extremely high-scaled expression values (» 2) were set to a maximum value of 2 without distorting the rest of the Figures. For dotplot visualization, first a euclidean distance matrix was generated for which hierarchical clustering was then applied. Ggtree was next used for dendrogram construction[101]. ReactomePA package was used for functional GSEA[102]. All unique detected genes in the cell subset were sorted by log fold change values to create ranked list that was inputted for GSEA analysis. enrichR was used to determine over and under expressed pathway from differential expression analysis (Kuleshov)[103]. Corrplot package was used for generation of correlation matrices. Volcano plots were constructed using EnhancedVolcano. Other graphical visualizations were created using ggplot2, ggpubr or plotly. All further downstream analysis was done in base R.

**Statistics and reproducibility**. Differential expression analysis of transcript abundance was assessed using Seurat's implementation of the nonparametric Wilcoxon rank-sum test. Genes were generally defined as statistically significant by Bonferroni adjusted $p$. value <0.05 and log fold change greater than 0.25. For NKT, a non-adjusted $p$. value was used to define differentially expressed genes due to very small sample size.

For comparison of cell-type proportion and MFI (either mean, median, or geometric median fluorescent intensity based upon the distribution) values, a two-tailed Student's $t$ test was performed to indicate statistical significance. Fisher's exact test was used for all comparisons of categorical variables in Supplementary Table 1. Additionally, Pearson correlation coefficient was used to indicate strength of measured correlations. For correlation statistics involving categorical values, categorical or factorial variables were converted to binary "dummy" variables (either 0 or 1) for the purpose of statistical calculations.

Given the tremendous degree of lymphopenia in patient samples, in flow cytometric analysis, samples in which the total cell count of a cell population of interest was critically low to the point where it was not comparable to the other samples in the dataset were systematically excluded from downstream analysis. For a given cell population in a dataset, a set cell number cutoff was determined and all samples with cell numbers below that cutoff were excluded. For data that was combined between different datasets, values were normalized by multiplication of a common factor to align the means between the different datasets. Additionally, in the BALF scRNA sequencing re-analysis, samples with too low T-cell count, healthy control 1 and severe 1, were excluded in accordance to guidance given by Liao et al. in the original paper.

All dots present on scatter plots and summary graphs are representative of a separate biological duplicate, indicative of a unique patient sample. Any technical duplicates found between datasets were averaged after normalization.

**Reporting summary**. Further information on research design is available in the Nature Portfolio Reporting Summary linked to this article.

## Data availability

Main source data used to generate figures were deposited to figshare and can be accessed under the accession number https://doi.org/10.6084/m9.figshare.22251151.

## Code availability

The source code used to reproduce our analysis can be accessed upon reasonable request from the corresponding author.

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

## Acknowledgements

The current study is conducted with the support from the University of Central Florida start-up funding and College of Medicine, University of Central Florida COVID-19 seed funding support to H.N. and from federal funds from the National Institutes of Health (NIH), Contract No. 75N9301900065 to A.S., D.W. We would like to thank Dr. Justine Tigno-Aranjuez for sharing flow antibodies. Reagents and protocol for the detection of COVID-19 antigen by ELISA were obtained through BEI Resources, NIAID, NIH.

## Author contributions

H.N. conceived the research project. S.G. and H.N. performed data analysis, drafted manuscripts, and wrote the paper. H.N., S.G., S.T.G., D.L., N.T.L., J.A., D.W., and A.S. contributed to revising the manuscript. H.N., S.G., A.A., K.E., D.L., T.H., S.A., and A.M. contributed to the experimental processes.

## Competing interests

UCF has filed for a patent of COVID-19 diagnostic assay based on current work. A.S. is a consultant for Gritstone Bio, Flow Pharma, Moderna, AstraZeneca, Qiagen, Fortress, Gilead, Sanofi, Merck, RiverVest, MedaCorp, Turnstone, NA Vaccine Institute, Emervax, Gerson Lehrman Group, and Guggenheim. L.J.I. has filed for patent protection for various aspects of T-cell epitope and vaccine design work.

## Ethic declaration

This study was approved by the IRB committee of AdventHealth hospital and UCF under protocols #1668907 and #1590483.
