## [Peer Review File · Communications Biology]

Reviewers' comments:

Reviewer #1 (Remarks to the Author):

In this study entitled "Metabolic dysregulation induces impaired lymphocyte memory formation during severe SARS-CoV-2 infection", Gurshanay et al claimed to assess metabolic properties of several subsets of leukocytes from COVID-19 patients. They reported that hypoxia and anaerobic glycolysis drive CD8, NK, NKT exhaustion, cause mitochondrial dysfunction in CD8, NK, NKT cells, and impair memory differentiation in CD8 and NK cells.

This is a highly descriptive study with no clear relevance. In addition, experiments are not performed to the highest standards (flow cytometry) or designed correctly (Direct analysis of metabolism by Seahorse technology is missing throughout the entire manuscript). I also do not understand the connection between metabolic exhaustion, elevated glycolysis and mitochondrial dysfunction. Same to conclude on the exhausted nature of a cell because of expression of a few mRNAs. In addition, it seems that the manuscript was written in a rush as not easily readable, the figures are poorly designed, and materials and methods are poorly written.

Some other main points:

1. Age and population demographics of control and patients cohorts are missing and brings into question whether the populations are comparable. A table describing demographic and clinical data of patients is missing, but is absolutely needed. Does disease severity impact metabolic phenotype differently? More information on the patients' disease and inflammatory status at the time-point of analysis would facilitate integration and interpretation of the results. Relevant information would be whether these patients were altogether in a similar stage of the disease, what time-point from diagnosis/onset of the symptoms/positive test the samples were taken or whether they had already developed a specific adaptive immune/antibody response. The authors group all covid-19 patients together, but do not show disease severity stratification.
2. What about gating strategy for flow cytometry? Which kind of controls did the authors used? Which antibodies did they used for staining? The list reported in Table S4 seems incomplete. Did they use FMO controls? The fact that all these information are missing brings into the question whether data are reproducible.
3. Authors claim that hypoxia affects the metabolic phenotype of immune cells. Yet the authors don't show this for their cohort. Therefore, the authors should start by showing this in their cohort of patients.
4. Authors claim that metabolic dysregulation affects immune cells. Supporting information regarding metabolic observation is practically devoid from any experimental observations made by the authors. They base this observation on the mere quantification of GLUT1, and in some cases, DBNG uptake. They do not measure glycolysis nor oxidative phosphorylation by using the Seahorse technology. They do not even mention FAO.
5. The term exhaustion is used throughout the entire manuscript in an improper way. The term "exhaustion" is not supported by the authors findings. Authors did not perform experiments aimed at quantifying functional properties of cells and cell proliferation.
6. The authors claim that mitochondrial dysfunctions are present in several subsets of immune cells, including CD8 T cells and NKT among others. Again, this claim is not supported by experimental evidences other than GSEA analysis. They do not measure/show: i) mitochondrial mass; ii) possible mitochondrial membrane depolarization; iii) oxygen consumption rate; iv) accumulation of dysfunctional mitochondria by using transmission electron microscopy.

7. Mitophagy is indicated as a keyword in this manuscript. Again, claims reporting "mitophagy" are not supported by data. To demonstrate that mitophagy is occurring in cells, gene expression is not sufficient. Protein levels of LC3-I and -II, PINK1, PARKIN, MFN2, FUNDC1, among others should be measured by western blot and the presence of autophagosomes containing mitochondria should be verified by transmission electron microscopy.

8. Correlations to the disease and inflammatory status of the patients at the time-point of analysis are not reported and this questions about the clinical relevance of the paper.

Reviewer #2 (Remarks to the Author):

The paper by Gurshaney et al. investigates PBMC from patients with COVID-19 infection for metabolic reprogramming in effector lymphocytes. In addition, the authors us published data to investigate these changes in the lung of COVID-19 patients. The authors claim that hypoxia and anaerobic glycolysis are responsible for lymphocyte dysfunction. This article deals with the of cause timely topic of immune reaction in COVID-19. However, the findings are somehow limited and there might be some over-interpretation in parts of the article.

Comments:

- The fact that scRNA-seq data is from published data is not properly addressed in several parts of the paper. Figure 1A should make clear that scRNA-seq data is from published datasets. The overview clearly needs to be revised. In addition, it is not clear whether the scRNA-seq data is from Zhang et al. or Liao et al., since both papers are cited, this need clarification in the text.
- The control group in the Liao et al. paper is a shortcoming of that paper, since this was obtained in a different setting and with different techniques. Can the authors confirm their findings using other datasets?
- What is the value of Fig 1G? In Fig 1H, there are too many clusters to use this color code.
- Is reduction in CD8 and NK cells due to inflammation per se or a typical finding of COVID-19?
- The paper implies that patients with metabolic syndrome and COVID-19 have an alternated immune response to SARS-CoV-2 in comparison to previously heathy persons with COVID-19. Are there any differences between these groups?
- The paper nicely shows correlation of marker expression with disease severity. However, the dysfunction itself is not addressed.

Reviewer #3 (Remarks to the Author):

The present manuscript by Gurshaney et al. tries to show that hypoxic condition during severe SARS-CoV-2 infection metabolically reprograms the effector lymphocytes to depend on glycolysis for their energy needs. The more dependence glycolysis (and less on fatty acid oxidation) leads to mitochondrial dysfunction, lymphocyte exhaustion, senescence, and hampers memory differentiation. Though several reports so far have been published regarding mitochondrial and T cell exhaustion in COVID 19 disease conditions, evaluating the metabolic phenotype of NKT cells is relatively new in this field. While mostly strong omics-based data acquisition and analysis are presented, a major deficiency of this study is that there is no functional data presented for any cellular subsets (T cells (CD4 or CD8), or NK or NKT). While dependence on glucose is necessary for effector T cells or NK cells function, the authors continue to stress on dysfunctional memory phenotype without showing any alteration in immediate effector functon(with specific cytokine response assays, degranulation, etc.). In absence of any antigen specificity and functional response data on the subsets discussed, the

argument about memory dysfunction comes across as very superficial, and such information is necessary to have any meaningful interpretation of the data that has been presented. It must also be noted that multiple papers have now established that even PD1+ T cells (viral-specific too) could exhibit potent effector cytokine and target cytotoxicity function. The expression of PD1 after the chronic infection has also been argued to limit pathology. Some examples are appended below: PMID: 23644506, PMID: PMC6112830, PMID: PMC7406116, PMID: 31591533.

Specific comments and concerns about the study:

1. It is now well established that T cells alter cellular metabolism during activation and clonal expansion. They switch from mitochondrial oxidative phosphorylation to glycolysis to meet their energy demands to differentiate into effector populations during acute infection. Several reports also show that exhausted T cells undergo metabolic reprogramming, including hampered glycolysis accompanied by dysregulated mitochondrial energetics. However, the authors correlated the glycolytic dependence of lymphocytes only with exhaustion phenotype. They might reconsider the premise of their study and perform experiments to distinguish between exhausted and effector phenotypes.
2. The authors here tried to correlate hypoxia with mitochondrial exhaustion. But did not give any evidence to show loss of mitochondrial mass and/or function. So providing microscopic image analysis/ extracellular flux analysis is necessary to prove their point.
3. The expression of surface markers and transcriptional programs of exhausted lymphocytes often overlap with activated cells' phenotype. In particular, since effector T cells transiently express most of the Inhibitory receptors during activation, only checking the expression of inhibitory receptors is insufficient to identify exhausted lymphocytes (mainly T cells). So, showing the evidence of impairment in effector cytokine production and proliferation might be needed side by side to decouple the effector and exhausted phenotype.
4. The authors attempted to demonstrate a diminished memory differentiation in response to SARS-COV2 infection. However, they used the samples taken only at the one-time point when the patients were hospitalized. A longitudinal study looking at the presence of T cells over an extended period would be more appropriate to follow the memory formation.
5. In the methodology section, the authors mentioned that 'Blood samples from hospitalized COVID-19 patients were collected from the Advent Health hospital....' But it is unclear what criteria they used to distinguish between the moderate and severe disease conditions (since all of them were hospitalized).

Rebuttal Letter

We are deeply appreciated for insightful comments from Reviewers. We have carefully addressed these comments in a point-by-point manner throughout the manuscript. The revision was highlighted in blue throughout the manuscript. Please see our answers as bellows.

Reviewer #1:

1. *A table describing demographic and clinical data of patients is missing.*

Response: We now provide detailed patient demographic and clinical data (see **Supplementary Table 1**) including:

1. Age
2. Gender
3. Ethnicity
4. Presence of prior metabolic/immune/etc. comorbidities
5. Presence of severe respiratory impairment
6. Presence of dysfunctional lung symptoms
7. Length of time between hospitalization and sample collection
8. Inflammatory status (elevated/normal c-reactive protein levels)
9. Supplemental oxygen

2. *What about gating strategy for flow cytometry?*

Response: We now provide a detailed gating strategy (**Supplementary Fig.1**) as well as representative histogram and flow plot throughout the manuscript. We also provide a comprehensive list of all antibodies used in current study (see **Supplementary Table 5**)

3. *Authors claim that hypoxia affects the metabolic phenotype of immune cells. Yet the authors do not show this for their cohort. Therefore, the authors should start by showing this in their cohort of patients.*

Response: We now provide comprehensive patient clinical and demographic information (**Supplementary Table 1**). We show that 28 out of the 52 COVID-19 patients in our cohort exhibited “severe respiratory impairment”, marked by the presence of either respiratory failure, hypoxemia, hypoxia, dependence on supplemental oxygen, dependence on mechanical ventilation, or acute respiratory distress syndrome. Additionally, 41 out of the 52 COVID-19 patients exhibited “dysfunctional lung symptoms” (please see **Methods** for comprehensive details on patient classification criteria). Thus, it is clear that lung dysfunction is heavily predominant in our cohort.

4. *Authors claim that metabolic dysregulation affects immune cells. Supporting information regarding metabolic observation is practically devoid from any experimental observations made by the authors. They base this observation on the mere quantification of GLUT1, and in some cases, 2-NBDG uptake. They do not measure glycolysis nor oxidative phosphorylation by using the Seahorse technology. They do not even mention FAO.*

Response: Due to the limited amount of blood that we received for each patient and the tremendous degree of lymphopenia, we were unable to sort the requisite number of cells to perform Seahorse assay. Instead, we now used single cell metabolomics assay (**SCENITH**) to comprehensively measure glycolytic flux, glucose dependence, and AAO and FAO capacities in small abundance subsets (see **Figs.2B,D, Fig.5B, Fig.6D**). Additionally, regarding FAO, we now also provide evidence of differential cpt1a expression (see **Fig.2F**).

5. *The term exhaustion is used throughout the entire manuscript in an improper way. The term "exhaustion" is not supported by the authors findings. Authors did not perform experiments aimed at quantifying functional properties of cells and cell proliferation.*

Response: We have now performed multiple *in vitro* assays to validate "exhaustion" on immune cell subsets (see **Fig.3**). In addition to increased cell surface marker expression of lag-3, we show that CD8⁺Tc from COVID-19 patients exhibit decreased proliferation (via Ki-67 expression), as well as decreased rate of cellular protein synthesis (required for the production of effector molecules and cytokine secretion). We also demonstrate a significantly decreased secretion of IFN γ amongst mitochondrially impaired CD8⁺Tc in COVID-19 patients.

6. *The authors claim that mitochondrial dysfunctions are present in several subsets of immune cells, including CD8 T cells and NKT among others. Again, this claim is not supported by experimental evidences other than GSEA analysis. They do not measure/show: i) mitochondrial mass; ii) possible mitochondrial membrane depolarization; iii) oxygen consumption rate; iv) accumulation of dysfunctional mitochondria by using transmission electron microscopy.*

Response: We now used single cell metabolomics assay (**SCENITH**) to comprehensively measure glycolytic flux, glucose dependence, and oxidation AAO and FAO capacities in various lymphocyte subsets (see **Figs.2B,D, Fig.5B, Fig.6D**). Specifically, we found that in response to oligomycin treatment, COVID(-) and healthy CD8⁺Tc exhibit greater reduction in cellular protein synthesis as compared to CD8⁺Tc from COVID-19 patients, indicating decreased mitochondrial flux in COVID-19 CD8⁺Tc. Additionally, we now also provide evidence that from COVID-19 (+) CD8⁺Tc have decreased mitochondrial membrane potential via TMRM staining (**Fig.2C, Fig.5E, Fig.9G**). We also show that cells with decreased mitochondrial mass in COVID-19 patients have impaired cytokine secretion (**Fig.3F**). Given that an accumulation of dead mitochondria has already been reported using electron microscopy (Siska et al., PID34779418), we cited this article and provide the discussion for the results we found.

7. ***Mitophagy is indicated as a keyword in this manuscript. Again, claims reporting "mitophagy" are not supported by data. To demonstrate that mitophagy is occurring in cells, gene expression is not sufficient. Protein levels of LC3-I and -II, PINK1, PARKIN, MFN2, FUNDC1, among others should be measured by western blot and the presence of autophagosomes containing mitochondria should be verified by transmission electron microscopy.***

Response: In addition to mRNA expression, we now also show an increase in the intracellular protein levels of LC3 in COVID-19 CD8⁺Tc. Additionally, to directly investigate the mitophagic dependence of immune cells in COVID-19, we leveraged mitophagy inhibitor Mdivi-1. We show that during SARS-CoV-2 specific peptide megapool stimulation of COVID-19 PBMCs, Mdivi-1 treatment strongly improves the activation, cytokine secretion, antigen-specific response, and metabolism of CD8⁺Tc, indicating a significant role of mitophagy in regulating the immunometabolism of CD8⁺Tc during COVID-19 (**Fig.8**).

8. *Correlations to the disease and inflammatory status of the patients at the time-point of analysis are not reported and this questions about the clinical relevance of the paper.*

Response: In our table of patient clinical and demographic information, we now provide information on whether the patients in our cohort had elevated c-reactive protein levels at the timepoint of analysis. We demonstrate correlations between the immunometabolic profiles of CD8⁺Tc and serum c-reactive protein, LDH, and glucose levels (see **Supplementary Fig.5**); however, due to the limited number of patients in our cohort, we were not able to perform such extensive correlation analysis.

Reviewer #2 (Remarks to the Author):

The paper by Gurshaney *et al.* investigates PBMC from patients with COVID-19 infection for metabolic reprogramming in effector lymphocytes. In addition, the authors us published data to investigate these changes in the lung of COVID-19 patients. The authors claim that hypoxia and anaerobic glycolysis are responsible for lymphocyte dysfunction. This article deals with the of cause timely topic of immune reaction in COVID-19. However, the findings are somehow limited and there might be some over-interpretation in parts of the article.

Response: We thank reviewer for recognizing the importance of this work.

1. *The fact that scRNA-seq data is from published data is not properly addressed in several parts of the paper. Figure 1A should make clear that scRNA-seq data is from published datasets. The overview clearly needs to be revised. In addition, it is not clear whether the scRNA-seq data is from Zhang et al. or Liao et al., since both papers are cited, this need clarification in the text.*

Response: We now clearly highlight in (**Fig.1A**) that the BALF data is from “Reanalysis of Single-Cell RNA Sequencing Data”. Additionally, the main BALF scRNA-seq dataset if from Liao *et al.*, not from Zhang *et al.* We have updated the references cited in the text to avoid the confusion. We only use one additional healthy control from Morse *et al.* as a reference suggested for use in the original paper by Liao *et al.*

2. *The control group in the Liao et al. paper is **a shortcoming of that paper**, since this was obtained in a different setting and with different techniques. Can the authors confirm their findings using other datasets?*

Response: We briefly performed analysis on a second PBMC scRNA sequencing dataset to confirm the observation in BALF dataset. The results are presented in **Supplementary Figs.6E-G**.

3. *What is the value of Fig 1G? In Fig 1H, there are too many clusters to use this color code.*

Response: We agree with reviewer’s concern. We have now removed the figure in question and have ensured that the new color scales used for cluster presentation are more easily distinguishable.

4. *Is reduction in CD8 and NK cells due to inflammation per se or a typical finding of COVID-19?*

Response: We thank Reviewer for this important comment. To confirm whether CD8 and NK lymphocytopenia is a hallmark of COVID-19, we compare the abundance of CD8 and NK in patients who have COVID-19 symptoms but are negative for COVID-19, COVID(-). We found that severe lymphocytopenia occurred is tremendously exacerbated in COVID-19 compared to COVID(-) patients (see **Fig.1**).

5. *The paper implies that patients with **metabolic syndrome** and **COVID-19** have an alternated immune response to SARS-CoV-2 in comparison to previously heathy persons with COVID-19. Are there any differences between these groups?*

Response: We now evaluate the immunometabolic distinction in CD8⁺Tc from COVID-19 patients with/without metabolic syndrome (see **Supplementary Fig. 5**); however, the limited number of patients is a challenge in our cohort.

6. *The paper nicely shows correlation of marker expression with disease severity. However, the dysfunction itself is not addressed.*

Response: Please see the response for question 4 of Reviewer 1.

Reviewer #3 (Remarks to the Author):

*The present manuscript by Gurshaney et al. tries to show that hypoxic condition during severe SARS-CoV-2 infection metabolically reprograms the effector lymphocytes to depend on glycolysis for their energy needs. The more dependence glycolysis (and less on fatty acid oxidation) leads to mitochondrial dysfunction, lymphocyte exhaustion, senescence, and hampers memory differentiation. Though several reports so far have been published regarding mitochondrial and T cell exhaustion in COVID 19 disease conditions, evaluating the metabolic **phenotype of NKT cells is relatively new in this field**. While mostly strong omics-based data acquisition and analysis are presented, **a major deficiency of this study is that there is no functional data presented for any cellular subsets** (T cells (CD4 or CD8), or NK or NKT). While dependence on glucose is necessary for effector T cells or NK cells function, the authors continue to stress on **dysfunctional memory phenotype without showing any alteration in immediate effector function** (with specific cytokine response assays, degranulation, etc.). In absence of any antigen specificity and functional response data on the subsets discussed, the argument about **memory dysfunction comes across as very superficial**, and such information is necessary to have any meaningful interpretation of the data that has been presented. It must also be noted that multiple papers have now established that even **PD1+ T cells** (viral-specific too) could exhibit potent effector cytokine and target cytotoxicity function. The expression of PD1 after the chronic infection has also been argued to limit pathology. Some examples are appended below: PMID: 23644506, PMCID: PMC6112830, PMCID: PMC7406116, PMID: 31591533.*

Response: We thank Reviewer for recognizing the novelty and importance of our work, and providing us with scientific advice and comprehensive references.

1. It is now well established that T cells alter cellular metabolism during activation and clonal expansion. They switch from mitochondrial oxidative phosphorylation to glycolysis to meet their energy demands to differentiate into effector populations during acute infection. Several reports also show that exhausted T cells undergo metabolic reprogramming, including hampered glycolysis accompanied by dysregulated mitochondrial energetics. However, the authors correlated the glycolytic dependence of lymphocytes only with exhaustion phenotype. They might reconsider the premise of their study and perform experiments to distinguish between exhausted and effector phenotypes.

Response: Using the SCENITH assay, we now directly validate the metabolism of exhausted LAG-3⁺ versus LAG-3⁻ CD8⁺Tc (see **Fig.6D**). In our metabolic readout, we also include analysis on non-COVID patients with COVID-19 like respiratory symptoms, highlighting that the difference in metabolism is not simply a result of increased activation due to infection, but are rather uniquely specific to COVID-19 (see **Figs.2B,D, Fig.5B, Fig.6D**).

2. The authors here tried to correlate hypoxia with mitochondrial exhaustion. But did not give any evidence to show loss of mitochondrial mass and/or function. So providing microscopic image analysis/extracellular flux analysis is necessary to prove their point.

Response: Please see the response to question 6 of Reviewer 1.

3. The expression of surface markers and transcriptional programs of exhausted lymphocytes often overlap with activated cells' phenotype. In particular, since effector T cells transiently express most of the Inhibitory receptors during activation, only checking the expression of inhibitory receptors is insufficient to identify exhausted lymphocytes (mainly T cells). So, showing the evidence of impairment in effector

cytokine production and proliferation might be needed side by side to decouple the effector and exhausted phenotype.

Response: Please see the response to question 5 of Reviewer 1.

*4. The authors attempted to demonstrate a diminished memory differentiation in response to SARS-COV2 infection. However, they used the samples taken only at the one-time point when the patients were hospitalized. A **longitudinal study** looking at the presence of T cells over an extended period would be more appropriate to follow the memory formation.*

Response: We now provide information regarding the time-point for sample collection (**Supplementary Table 1**). We agree that a longitudinal study will support the conclusion on memory formation. While other group (Sarah Adamo et al, Nature 2022) already study that, we assessed PBMCs from COVID-19 patients that were collected on average around 14 days after the onset of hospitalization, which allows us to focus on the interphase of primary response and secondary response. This is an ultimate important point of our current study.

*5. In the methodology section, the authors mentioned that 'Blood samples from hospitalized COVID-19 patients were collected from the Advent Health hospital....' But it is unclear what criteria they used to distinguish between the **moderate and severe disease** conditions (since all of them were hospitalized).*

Response: For the BALF and PBMC scRNA sequencing reanalysis, the criteria used to differentiate between moderate and severe disease by the original authors have been listed in the methods. Regarding the PBMCs used for high-dimensional flow cytometry and metabolomics analysis, we now provide detailed patient clinical and demographic information for our cohort in **Supplementary Table 1**. However, due to the low amount of blood that we received for each patient and the tremendous degree of lymphopenia, different samples amongst the total cohort had to be distributed towards different experiments. Thus, combined with the relatively lower number of patients in our overall cohort, it was not statistically feasible to stratify hospitalized patients into moderate and severe disease for analysis.

REVIEWERS' COMMENTS:

Reviewer #1 (Remarks to the Author):

The authors addressed all issue raised by the reviewer.

Reviewer #2 (Remarks to the Author):

From my point of view, the authors provide a sufficient version revision of their paper. I only have on minor comment concerning the gating strategy: the CD8 plot is shown twice, there might be a way to avoid this.